# Sequence and structural determinants of RNAPII CTD phase-separation and phosphorylation by CDK7

Katerina Linhartova [1,2], Francesco Luca Falginella[1,3], Martin Matl [1,3], Marek Sebesta [1] ✉, Robert Vácha [1] ✉ & Richard Stefl [1,2] ✉

The intrinsically disordered carboxy-terminal domain (CTD) of the largest subunit of RNA Polymerase II (RNAPII) consists of multiple tandem repeats of the consensus heptapeptide Y1-S2-P3-T4-S5-P6-S7. The CTD promotes liquid-liquid phase-separation (LLPS) of RNAPII in vivo. However, understanding the role of the conserved heptad residues in LLPS is hampered by the lack of direct biochemical characterization of the CTD. Here, we generated a systematic array of CTD variants to unravel the sequence-encoded molecular grammar underlying the LLPS of the human CTD. Using in vitro experiments and molecular dynamics simulations, we report that the aromaticity of tyrosine and *cis-trans* isomerization of prolines govern CTD phase-separation. The *cis* conformation of prolines and β-turns in the SPXX motif contribute to a more compact CTD ensemble, enhancing interactions among CTD residues. We further demonstrate that prolines and tyrosine in the CTD consensus sequence are required for phosphorylation by Cyclin-dependent kinase 7 (CDK7). Under phase-separation conditions, CDK7 associates with the surface of the CTD droplets, drastically accelerating phosphorylation and promoting the release of hyperphosphorylated CTD from the droplets. Our results highlight the importance of conformationally restricted local structures within *spacer* regions, separating uniformly spaced tyrosine *stickers* of the CTD heptads, which are required for CTD phase-separation.

RNA polymerase II (RNAPII) primarily transcribes protein-coding genes[1]. The largest subunit of RNAPII—RPB1—contains a low-complexity, intrinsically disordered carboxy-terminal domain (CTD), which forms a tail-like extension from the catalytic core of RNAPII[1]. The CTD is conserved from yeast to humans and consists of a repeated heptapeptide motif with the consensus sequence Y1-S2-P3-T4-S5-P6-S7 (Fig. 1a, Supplementary Fig. 1a)[2]. However, CTDs vary in both length and the level of degeneracy of the heptapeptide motif across eucaryotes[3]. The CTD is targeted by CTD kinases and can be phosphorylated on Y1, S2, T4, S5, and S7 of the heptapeptide motif,

respectively. The phosphorylation pattern of the CTD varies throughout the transcription cycle, orchestrating the recruitment of specific factors. Association of the CTD phosphorylation patterns with particular events in the transcription cycle gave rise to the so-called "CTD code" model[4–9]. The CTD is mostly unstructured in solution with a residual β-turn structure, but it could become ordered upon binding by transcription and/or RNA processing factors[4,6,7,10–13].

Whilst five of the seven residues in the CTD heptapeptide motif can be phosphorylated and their importance for transcription regulation has been established[14–20], the functional significance of the

---

[1]CEITEC - Central European Institute of Technology, Masaryk University, Brno, Czechia. [2]National Centre for Biomolecular Research, Faculty of Science, Masaryk University, Brno, Czechia. [3]These authors contributed equally: Francesco Luca Falginella, Martin Matl. ✉e-mail: marek.sebesta@ceitec.muni.cz; robert.vacha@ceitec.muni.cz; richard.stefl@ceitec.muni.cz

**Fig. 1 | The importance of human RNAPII C-terminal domain (CTD) residues in phase-separation. a** A schematic representation of transcribing RNAPII highlighting the CTD of the largest subunit, RPB1, and the CTD heptad consensus sequence (*left*). Sequence logo for the human CTD heptad (*right*), *n* = number of heptad repeats. **b** A schematic representation of *cis-trans* prolyl-peptidyl isomerization of proline and hydroxyproline residues, respectively, (*left*) along with their free energy profiles (*right*). **c** Simplified sequences of the heptad repeat of the CTD variants used in this study. **d** Quantification of the in vitro liquid-liquid phase-separation (LLPS) assay, performed with the mGFP-CTD and its tyrosine variants, mGFP-CTD[YIF] and mGFP-CTD[Y1A]. Each dot represents a detected droplet, the black dots indicate the median of droplet size per measurement (*n* = 3). The black line and error bars represent the mean ± SD of the three medians. **e** Same as (**d**) but with the mGFP-CTD and its proline variants, mGFP-CTD[P3G] and mGFP-CTD[Hyp]. **f** Representative micrographs of three independent LLPS mixing assays with mGFP-

CTD (green) and mCherry-CTD (red), and with mGFP-CTD[Hyp](green) and mCherry-CTD (red), respectively. Merge = overlay of mCherry and GFP channels. Scale bar = 10 µm. (*) The intensity of all micrographs was uniformly enhanced in Fiji[125] for better visibility. **g** The quantification of LLPS mixing assays shown in (**f**). Each dot represents a detected droplet, black dots indicate the median of droplet size (*left*), integrated intensity of all droplets from the mCherry (*middle*) or GFP (*right*) channels per measurement (*n* = 3). The black line and the error bar represent the mean ± SD of the three medians. Statistical comparison of means in datasets was determined via an unpaired, two-sided *t*-test (see "Methods" FM-image analysis), only *p* ≤ 0.05 is depicted. *p*-values: ((**g**) middle) * = 0.026, ((**g**) right) * = 0.019, ** = 0.0062, *** = 0.0009, respectively. **h** Same as (**d**) but with the mGFP-CTD and its serine variants, mGFP-CTD[S2A], mGFP-CTD[S5A], and mGFP-CTD[S7A]. **i** Same as (**d**) but with the mGFP-CTD and its threonine variant, mGFP-CTD[T4G]. All assays presented here were done in triplicates. Source data are provided as a Source Data file.

conserved prolines (P3 and P6) remains poorly understood[4]. Proline's pyrrolidine ring restricts proline conformational dynamics, enabling this amino acid to form two slowly interconverting *cis* and *trans* isomers[21]. The *cis* and *trans* proline conformers in the CTD heptapeptide motif create binding sites for CTD interacting factors, including CTD modifying enzymes and the CTD code readers[4,13,22,23]. The intrinsically slow interconversion of the two isomers, which is further slowed down if the serine or threonine preceding the proline is phosphorylated[24], can be accelerated by the conserved group of peptidyl-prolyl *cis-trans* isomerase enzymes (PPIases)[25]. PPIases are cotranslational chaperones, which assist in protein folding and in regulation of disease-associated misfolded intrinsically disordered proteins (IDPs)[26,27]. Two enzymes have been reported, Ess1 in yeast and PIN1 in mammals, which can catalyze transition between the two isomers of P3 and/or P6 in the phosphorylated CTD heptapeptide motif[28,29]. Prolines therefore may serve as a fundamental molecular switch used for the controlled timing of key biological processes (Fig. 1b)[30–32].

Recent data have indicated that the CTD can undergo cooperative liquid-liquid phase-separation in vitro and that the CTD is critical for the formation of hubs of RNAPII in human cells[33]. Interestingly, these RNAPII hubs are formed through CTD–CTD interactions when the CTD is not modified by phosphorylation[33]. Model of condensate-based transcriptional organization has emerged from recent studies[34], suggesting that the transcribing RNA polymerase II (RNAPII) may cluster in condensates associated with transcription initiation and elongation in a phosphorylation-dependent manner[35–38].

The low complexity and intrinsically disordered nature of the CTD provide RNAPII with molecular features that support phase-separation[39]. The highly repetitive nature and uniform patterning of the tyrosine residues in the CTD sequence align with the *sticker*-and-*spacer* model used in phase-separation of IDPs and multidomain proteins[40,41]. In this model, the phase-separation is driven by weak intermolecular interactions among *stickers*, whilst *spacers* play a role in either facilitating or inhibiting the formation of these interactions. The separation of *stickers* prevents them from engaging in overly strong gel or solid-inducing interactions and maintains the unstructured state of IDPs[42–44]. However, the conserved nature of the CTD suggests that the spacer residues in CTD might have further roles beyond serving as passive spacers. Furthermore, recent findings suggest that the mechanisms driving phase-separations are multifaceted, influenced not only by the identity of individual amino acids but also by the context of their surrounding sequence[45].

Here, we investigate the role of CTD residues that drive its phase-separation, control CTD's structural properties, and are required for phosphorylation by CTD kinases using in vitro and in silico approaches. Our work focuses on the role of conserved CTD residues, which act as *spacers* in the *sticker*-and-*spacer* phase-separation model, with the emphasis on proline residues and their *cis-trans* isomerization. Additionally, we investigate the phosphorylation of CTD by the CDK7 complex in solution and under phase-separated conditions, comparing the kinetics and spatial organization of the process.

## Results

### Preparation of RNAPII CTD variants

To decipher the sequence features of the CTD that facilitate its phase-separation, we aimed to generate single-point variants of all amino acids, two- and three-serine variants, and two-proline variant of the human CTD heptapeptide motif within all 52 repeats (Supplementary Table 1). We successfully cloned and expressed mGFP-CTD and its 13 variants (Fig. 1c and Supplementary Fig. 1b, and Supplementary Table 1). However, two proline variants (mGFP-CTD$^{P6G}$ and mGFP-CTD$^{P3,6G}$; glycine also disfavors formation of secondary structures[46]) yielded only insoluble material, which could not be refolded despite considerable effort (Supplementary Table 1). The insolubility is in line

with AlphaFold2[47] models in which the substitution of proline at position 6 (mGFP-CTD$^{P6G}$) and at positions 3 and 6 (mGFP-CTD$^{P3,6G}$), respectively, impose a repetitive cross-β structure typical of amyloid fibrils in the CTD models, predicted with high confidence (Supplementary Fig. 1c)[48,49]. These structures are in strong contrast with the wild-type CTD, which is predicted to be disordered in the mGFP-CTD model.

The challenge in creating the mGFP-CTD$^{P6G}$ and mGFP-CTD$^{P3,6G}$ variants led us to explore an alternative approach. We used a proline analogue, *trans*-4-hydroxy-L-proline (hydroxyproline; Hyp), to alter the typical *cis-trans* isomerization of prolines at positions 3 and 6 in the CTD heptapeptide motif. The inductive effect of the hydroxyl group at position 4 of the pyrrolidine ring reduces the energetic barrier to *cis-trans* isomerization and alters the prolyl peptide bond equilibrium constant, favoring the *trans* conformation (Fig. 1b)[50–52]. Using a proline-auxotrophic *Escherichia coli (E. coli)* strain[53] and Hyp as a proline analog, we generated a human CTD variant with prolines substituted with hydroxyprolines, tagged with monomeric green fluorescent protein mGFP-CTD$^{Hyp}$. Trace amounts of proline in the hydroxyproline source resulted in a minor byproduct with low hydroxyproline content, which could not be separated from the mGFP-CTD$^{Hyp}$ (Supplementary Fig. 1b).

### Tyrosine and proline residues drive CTD phase-separation

To reveal the precise role of the CTD heptad residues in phase-separation, we performed in vitro liquid-liquid phase-separation (LLPS) assays with human mGFP-CTD, its variants, and mCherry-CTD. We observed that mGFP-CTD formed spherical droplets in a protein concentration-dependent manner using a buffer containing 10 % dextran, as molecular crowding agent (Fig. 1d and Supplementary Fig. 1d). We note that although RNAPII concentration in the nucleus is estimated to be ~1 µM, it often increases locally by several orders of magnitude[54]. The obtained droplets were sensitive to hexane-1,6-diol (Supplementary Fig. 1d), confirming the liquid-like character of these droplets. We also tested whether tagging of CTD with mGFP and mCherry, respectively, affects CTD's ability to undergo LLPS and whether the tags themselves can phase-separate under conditions used in the assays. We found that the tags alone did not phase-separate (Supplementary Fig. 1e) and that the differences in LLPS properties between the tagged and non-tagged CTD were negligible. (Supplementary Fig. 1f-i).

We observed phase-separation of the mGFP-CTD$^{Y1F}$ variant, albeit less efficient, and the droplets were smaller compared to the wild-type CTD. The substitution of Y1 to alanine (mGFP-CTD$^{Y1A}$), however, completely abrogated observable phase-separation (Fig. 1d and Supplementary Fig. 2a), even at concentrations as high as 45 µM (Supplementary Fig. 2b).

Next, we tested the effect of the proline variants on CTD phase-separation. The mGFP-CTD$^{P3G}$ variant showed diminished ability to undergo LLPS compared to mGFP-CTD (Fig. 1e and Supplementary Fig. 2c), whereas the mGFP-CTD$^{P6G}$ and mGFP-CTD$^{P3,6G}$ variants could not be tested due to their insolubility (Supplementary Fig. 1c). Alternatively, we used the proline-to-hydroxyproline (mGFP-CTD$^{Hyp}$) variant, which had a significant impact on CTD phase-separation as it did not phase-separate within the studied concentration range from 2.5 to 10 µM (Fig. 1e and Supplementary Fig. 2c). However, the mGFP-CTD$^{Hyp}$ variant did phase-separate at concentrations of 15 µM and higher. (Supplementary Fig. 2d). The phase-separation observed at higher mGFP-CTD$^{Hyp}$ concentrations may be due to the presence of a byproduct with low hydroxyproline content in our sample (Supplementary Fig. 1b), which could still undergo phase-separation. Additionally, time-course mixing experiment with the wild-type CTD and the hydroxyproline variant (mGFP-CTD$^{Hyp}$) demonstrated that whilst the mGFP-CTD fused easily with the pre-formed droplets of mCherry-CTD in a time-dependent manner, the mGFP-CTD$^{Hyp}$ variant did not integrate

into the pre-existing mCherry-CTD droplets (Fig. 1f, g and Supplementary Fig. 2e).

We subsequently tested whether an enhanced rate of proline *cis-trans* interconversion, facilitated by specific isomerases, would influence CTD phase-separation. To this end, we used the isomerases PPIA and PIN1 to assess their impact on the phase-separation properties of the CTD. The PPIA isomerase is a single-domain enzyme with a broad specificity for proline-containing substrates[55]. Interestingly, we found that both PPIA and its catalytically inactive variant (PPIA[R55A])[56] dissolved preformed CTD droplets in a concentration-dependent fashion, suggesting that PPIA reverses CTD phase-separation through direct binding, independently of its PPIase activity (Supplementary Fig. 3a). The droplets dissolved more effectively with PPIA[R55A], likely due to its higher binding affinity and/or slower dissociation rates towards the CTD compared to the wild-type PPIA.

To test the importance of interaction between the isomerase and CTD we used PIN1, which contains an extra WW domain in addition to the catalytic PPIase domain, which provides binding affinity for its primary phosphoserine-proline substrates. Given that the WW domain has undetectable binding affinity towards non-phosphorylated CTD[57], it served as a negative control. We found that PIN1 and its catalytic variant (PIN1[C113S])[58–60] had no impact on the preformed CTD droplets, even when added at concentrations up to 160 µM (corresponding to 0.6:1 ratio relative to the CTD, when the CTD concentration is expressed in terms of heptad repeats (Supplementary Fig. 3b)). This observation supports the notion that the interaction with the CTD rather than the enzymatic activity may be responsible for the observed dissolution of the preformed droplets by PPIA.

Altogether, our experiments suggest that the aromaticity of the Y1 residue and possibly the *cis-trans* isomerization of prolines are key for CTD phase-separation.

## Threonine and serine residues modulate CTD droplet morphology

We further investigated the role of the serine and threonine residues in the CTD heptapeptide motif and used the mGFP-CTD[S2A], mGFP-CTD[S5A], mGFP-CTD[S7A], and mGFP-CTD[T4G] variants, respectively (Fig. 1h, i and Supplementary Fig. 2f, g). We observed that the mGFP-CTD[T4G] variant displayed reduced ability to form droplets compared to the wild-type CTD. Serine substitutions in the CTD heptapeptide motif drove phase-separation, as observed for the wild-type CTD, but the droplets differed in size and number (Fig. 1h). In particular, the droplets formed by the mGFP-CTD[S2A] and mGFP-CTD[S5A] variants were smaller and more abundant compared to the wild-type and mGFP-CTD[S7A]. The position-dependency of serine substitutions on droplet size and number was further confirmed through the characterization of variants in which two (mGFP-CTD[S2,5A], mGFP-CTD[S2,7A], and mGFP-CTD[S5,7A]) and all three serine residues were substituted for alanine residues (mGFP-CTD[S2,5,7A]) (Supplementary Fig. 4a–d). Interestingly, we observed that the mGFP-CTD[S2,5A] and mGFP-CTD[S2,5,7A] variants formed not only spherical droplets but also a notable number of small, non-spherical aggregates, which were both dissolved upon the addition of hexane-1,6-diol (Supplementary Fig. 4e, f). We note that these non-spherical aggregates were discarded during our image analysis due to their shape (Supplementary Fig. 4f). This evidence suggests that the properties of serine residues are important to attenuate the strong tyrosine interactions preventing aggregation and maintaining conditions optimal for LLPS. In addition, serine and threonine substitutions may perturb the β-turn propensity of the SPXX motifs in the heptad repeat and thereby potentially affect the material properties of the droplets.

## Impact of CTD heptad residues on LLPS: insights from MD simulations

To investigate the impact of the individual CTD substitutions on molecular interactions and the characteristics of phase-separation, we conducted coarse-grained (CG) and all-atom molecular dynamics (MD) simulations with explicit solvent (Supplementary Table 3). Consistent with our experimental LLPS assays, the analysis of the intramolecular distances in full-length (i.e., 52 heptad repeats) single-chain CG simulations (Fig. 2a) revealed that the CTD[Y1A] variant, unlike the CTD[Y1F] variant, adopted a less compact conformation compared to CTD[cons] (which includes only the heptad consensus sequence). The critical role of an aromatic residue at position 1 of the heptad repeat in CTD phase-separation was further emphasized in our condensed phase CG simulations of full-length CTD molecules. We found that the CTD[Y1A] variant, unlike CTD[cons] and the CTD[Y1F] variant, not only inhibited spontaneous condensate formation but also dissolved preformed condensates within the explored time scale (Fig. 2b and Supplementary Fig. 5a, b). Notably, our condensed phase CG simulations revealed enhanced interactions of phenylalanine residues (Supplementary Fig. 5c), which led to more compact condensates for the CTD[Y1F] variant, as indicated by the size distribution of the protein clusters (Supplementary Table 4). In line with phase-separation experiments, in all-atom simulations with two di-heptads of CTD[cons] or the variants CTD[Y1F] and CTD[Y1A], the analysis of three independent replicas demonstrated that the strength of the side chain interaction energy follows the order Y > F > A. Notably, for all three constructs we observed a considerable energetic contribution originating from backbone-side chain interactions (Supplementary Fig. 5d).

To test the critical role of *cis-trans* proline isomerization on LLPS, we performed additional all-atom MD simulations of CTD[cons] and its hydroxyproline variant (CTD[consHYP]). Specifically, we explored combinations of the isomerization state of prolines and hydroxyprolines at positions 3 and 6 of the heptad repeat: *trans-trans* (i.e., major state), *cis-trans*, *trans-cis*, and *cis-cis* (note that only the first and last combinations were included for CTD[consHYP]). Overall, we observed that all *cis* isomers showed greater compactness (Fig. 2c and Supplementary Figs. 6a and 7a), a smaller radius of gyration (Fig. 2d and Supplementary Figs. 6b and 7b), and a general increase of the per-residue intramolecular interaction energies across the entire CTD di-heptad (Fig. 2e and Supplementary Figs. 6c and 7d). This behavior was particularly observed in the case of the double *cis* isomers (CTD[cisP3,6] and CTD[cisHYP3,6]). Interestingly, both CTD[consHYP] and CTD[cisHYP3,6] variants showed minor shifts of conformational populations towards more compact structures when compared with the proline counterparts. This shift likely originates from the additional local interactions involving the extra OH group of hydroxyproline (Supplementary Figs. 7d and 8a). However, the shift from the *cis* isomer is significantly larger.

Next, we assessed the condensation behavior of the CTD[S2A], CTD[T4G], and CTD[S7A] variants. In single chain CG simulations, all three variants exhibited less compact conformations compared to CTD[cons] (Supplementary Fig. 9a), reflecting a reduced propensity for phase-separation. This observation was corroborated in condensed phase CG simulations, which showed poor protein cluster formation for these variants within the explored time scale (Supplementary Fig. 9b, c). The low resolution of the used CG model precludes a detailed analysis of secondary structure formation. Therefore, to investigate the role on CTD phase-separation of β-turn propensity within the SPXX motifs in the *spacer* regions, we employed atomistic MD simulations. For each variant, including CTD[cons], we observed bimodal distributions of the distance between Cα atoms of the residues at positions 1 and 4, typically used to characterize β-turns (Fig. 2f). Two populations were identified: *compact* and *extended*, demarcated by the 0.7 nm cutoff. The serine variants, particularly CTD[S2A], destabilized the β-turns, favoring more extended conformations, compared to CTD[cons]. This destabilization stems from the loss of serine hydroxyl group, which significantly altered both the overall local and SP-turn specific hydrogen bonding pattern[61] (Supplementary Fig. 8b). In contrast, the CTD[T4G] variant showed a considerable increase in the *compact* population,

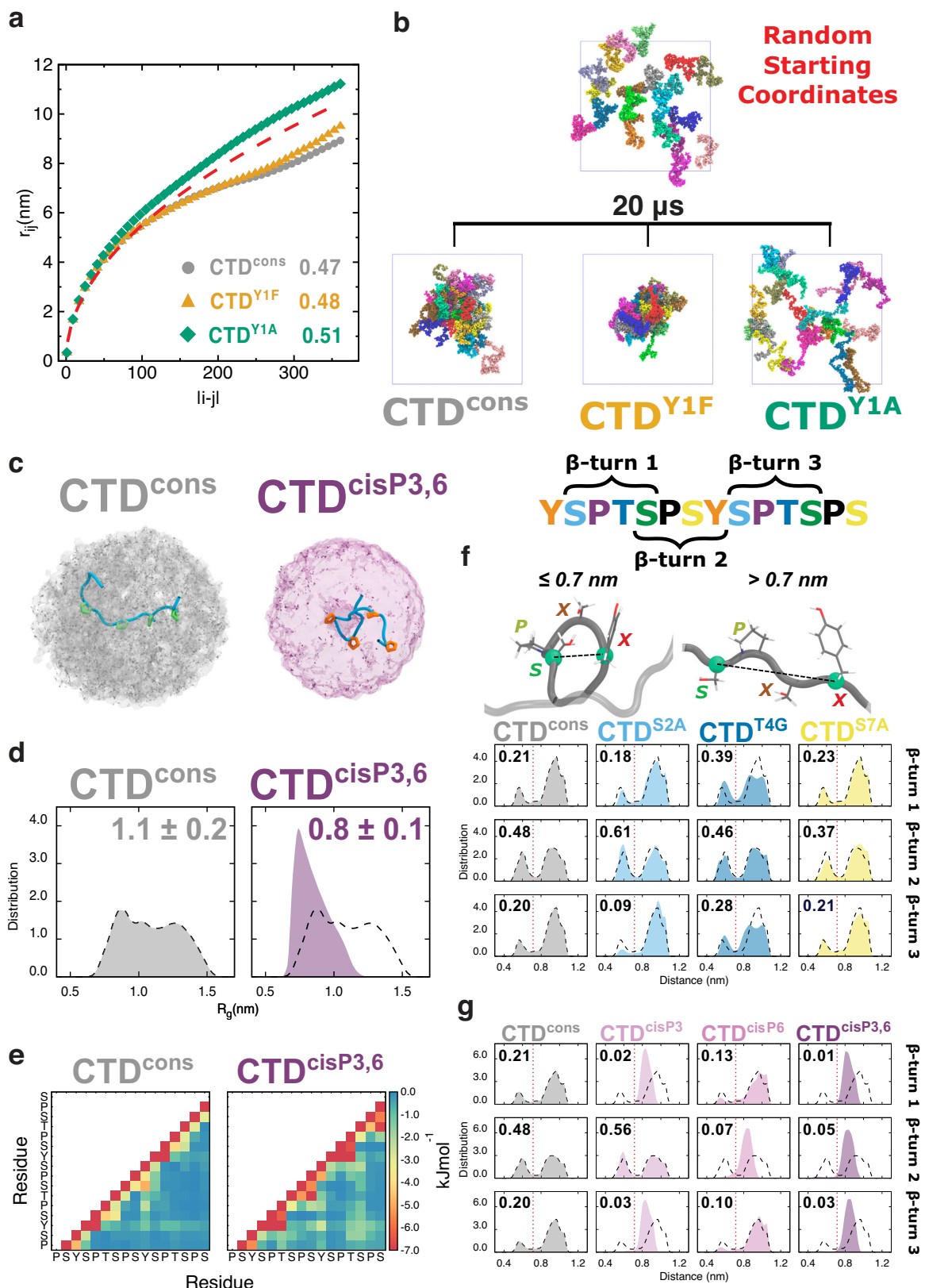

consistent with the stabilizing effect of glycine on β-turn structures[62]. Interestingly, the Cα-Cα distance patterns in the (hydroxy)proline isomerization variants exhibited significant deviations. The previously observed bimodal distributions were replaced by narrower, single-peaked distributions in each β-turn with a *cis* (hydroxy)proline (Fig. 2g and Supplementary Fig. 7c). We emphasize that, although the

additional interactions of hydroxyproline can introduce local perturbations in the Cα-Cα distance pattern (*i.e.*, small shoulders in the single-peaked distributions), the overall behavior associated with the *cis* isomerization state is maintained. Altogether, we conclude that the structural properties within the *spacer* region of the CTD heptad play an important role in either facilitating or suppressing the intra-

**Fig. 2 | MD simulations of RNAPII CTD constructs. a** Average intramolecular distance ($r_{ij}$) as a function of residue separation (|i-j|) for CTD$^{cons}$ (gray circles), variants CTD$^{YIF}$ (orange triangles), and CTD$^{YIA}$ (green diamonds) in CG single chain simulations. The values of the Flory scaling exponent ($v$; see "Methods" for details) are also shown (maximum SD is 0.003). The red dashed line represents the scaling behavior of an ideal chain. **b** Representative snapshots from CG condensed phase simulations of CTD$^{cons}$, variants CTD$^{YIF}$ and CTD$^{YIA}$. Each molecule is colored differently using van der Waals spheres. Solvent omitted for clarity. **c** Density maps of the carbon atoms in the N- and C-termini capping groups for CTD$^{cons}$ (*left*) and the double *cis* proline variant CTD$^{cisP3,6}$ (*right*). Two representative conformations of di-heptad constructs are shown (light blue tubes) with *trans* (green) and *cis* (orange) prolines. **d** Distributions of the radius of gyration for the constructs in (**c**). Mean ± SD is shown. **e** Per-residue intramolecular interaction energy maps for the

constructs in (**c**). Values matching or exceeding −7.0 kJ/mol are colored in dark red. SD (0.003-2.586). **f** Representative structures of a compact (*top left*) and extended (*top right*) SPXX-motif, as defined by the distance (dashed line) between the Cα atoms of residues at positions 1 and 4 (green spheres). Backbone, tube; atoms, sticks. The distributions of the Cα-Cα distances for the CTD$^{cons}$, CTD$^{S2A}$, CTD$^{T4G}$, and CTD$^{S7A}$ variants are also shown. The three SPXX motifs within a single di-heptad are displayed separately from *top* to *bottom*. In each plot, we provide the ratio between the area under the curve (AUC) up to 0.7 nm cutoff value (red dashed line) and the AUC beyond the cutoff. **g** Distributions of the Cα-Cα distances same as in (**f**) for CTD$^{cons}$ and all *cis* proline variants. Note that shown results from all-atom simulations (**c**–**g**) were averaged over three replicas and all di-heptad constructs. In (**d**, **f**, and **g**) the distributions for CTD$^{cons}$ are reprised as dashed black lines for ease of comparison. Source data are provided as a Source Data file.

and inter-residual interactions among CTD *stickers*, thereby impacting its phase-separation properties.

## The CDK7 complex associates with the surface of CTD droplets and reverses phase-separation by CTD phosphorylation

It has been shown that CTD phosphorylation by the kinase module of the transcription initiation factor TFIIH, the Cyclin-dependent kinase 7 (CDK7) complex releases RNAPII from the phase-separated hubs[33,38]. With the array of CTD variants at hand, we initially tested the role of the specific residues within the CTD heptad consensus in kinase assays with the CDK7 complex, comprising CDK7, MAT1, and Cyclin H. The initial reaction velocities measured in the kinase assay, conducted with the CDK7 complex and the CTD variants, pinpointed the complete consensus sequence which is required for CTD phosphorylation (Fig. 3a and Supplementary Fig. 10a–e). We observed that the CTD variants containing substitutions at specific positions−Y1A, Y1F, P3G, and Hyp−were not phosphorylated by the CDK7 complex. Additionally, our results showed that the phosphorylation was partially impaired when the T4G variant was used. Interestingly, we observed that CTD phosphorylation by the CDK7 complex did not occur when the S5 residue was substituted with alanine. This suggests that the minor phosphorylation site at the S7 residue, which is observed in in vitro CTD phosphorylation by the CDK7 complex, likely requires a pre-phosphorylated S5 residue. Additionally, we observed increased phosphorylation of the S2A variant compared to the WT. A recent crystal structure of the activated CDK7 complex with a modeled CTD peptide substrate[63], extrapolated from the Cdk2/CycA/substrate complex structure[64], suggests that the S2A substitution may introduce additional favorable hydrophobic interactions in the binding pocket, likely making S2A a more effective substrate in our in vitro assay.

In summary, our data suggest that the full consensus sequence for phosphorylation at S5 of the CTD is (YXPXSPX)Y, with the tyrosine residue being either within the phosphorylated heptad or in the following one. This consensus aligns with the recent modeling of the CTD substrate into the crystal structure of the human-activated CDK7 complex.

To reveal molecular insights into the CTD droplet dissolution facilitated by the CDK7 complex phosphorylation, we incubated the preformed mGFP-CTD droplets with the CDK7 complex in the presence and absence of adenosine triphosphate (ATP), respectively. In the presence of ATP, the droplets exhibited a gradual reduction in size and number, with increasing time and the CDK7 complex concentrations, undergoing complete dissolution after a 10-minute incubation at concentration of 1 μM CDK7 (Fig. 3b, c and Supplementary Fig. 11a–c). No droplet dissolution occurred in the absence of ATP. We observed only small differences when we induced the mGFP-CTD phase-separation before adding the CDK7 complex (Supplementary Fig. 12a, b), compared to the scenario in which we mixed the CDK7 complex with mGFP-CTD before inducing phase-separation. In the latter experiment, smaller droplets formed, and this effect was more pronounced with increasing concentration of CDK7 (Supplementary

Fig. 12c, d). Furthermore, the CDK7 complex alone does not undergo phase-separation (Supplementary Fig. 12e) nor did we observe effect of 0.5 mM ATP on the mGFP-CTD droplet formation (Supplementary Fig. 12f, g). In agreement with these experimental observations, performed condensates of CTD$^{cons}$ dissolved upon phosphorylation of serine at position 5 or 7 in condensed phase CG simulations (Fig. 3d and Supplementary Fig. 13a). Similarly, in all-atom simulations, the electrostatic repulsion between the negatively charged phosphor-residues destabilized the intramolecular interactions within the CTD di-heptads, resulting in more expanded conformations (Supplementary Fig. 13b, c).

Subsequently, we tested whether the CDK7 complex is recruited into the droplets where it could introduce multiple negative charges on the CTD, destabilizing the droplets[65]. In sedimentation assay, we found that the CDK7 complex is enriched in the pelleted fraction containing droplets of the CTD. This observation suggests co-localization of the CDK7 complex with CTD droplets (Fig. 3e). To visualize the co-localization of the CDK7 complex with the CTD droplets, we used both wide-field and super-resolution imaging techniques. Wide-field microscopy indicated that the CDK7 complex, labeled with Alexa594, appeared to localize at the surface of the CTD droplets (Fig. 3f and Supplementary Fig. 11d). This was further corroborated by super-resolution imaging, which revealed a thin layer of the CDK7 complex enveloping the CTD droplet (Fig. 3g). The CDK7 complex remained localized at the surface of the droplets also when it was preincubated with mGFP-CTD before inducing the phase-separation (Supplementary Fig. 11e).

Altogether, this suggests that the CDK7 complex associates with the surface of CTD droplets where it likely phosphorylates available polar hydroxyl groups of residues S5 and to a lesser extent S7[66] in a distributive manner (Supplementary Fig. 12h).

## Phase-separation appears to accelerate CTD phosphorylation

Next, we wondered whether CTD phosphorylation is accelerated under phase-separation conditions when the CDK7 complex associates with the surface of CTD droplets. To test this, we incubated the CTD substrate at concentrations below (0.25 and 0.5 μM) and above (2.5 and 5 μM) the threshold at which droplet formation was observed in fluorescence microscopy (Fig. 1), with the CDK7 complex and ATP, in the presence and absence of dextran, respectively, which facilitates phase-separation in vitro. We found that CTD phosphorylation by the CDK7 complex is accelerated only when dextran is present and the CTD substrate is at higher concentrations (Fig. 3h, Supplementary Fig. 11f, g), which are the conditions that showed a robust droplet formation in our imaging and sedimentation assays (Fig. 1 and Supplementary Fig. 12i–k)

To rule out the possibility that the presence of dextran in the solute phase affects phosphorylation, we performed a sedimentation kinase assay[67]. This approach enables the separation of enzymatic activities in different phases (Fig. 3i). We observed that phosphorylation rates in the bulk (a supernatant fraction after centrifugation with

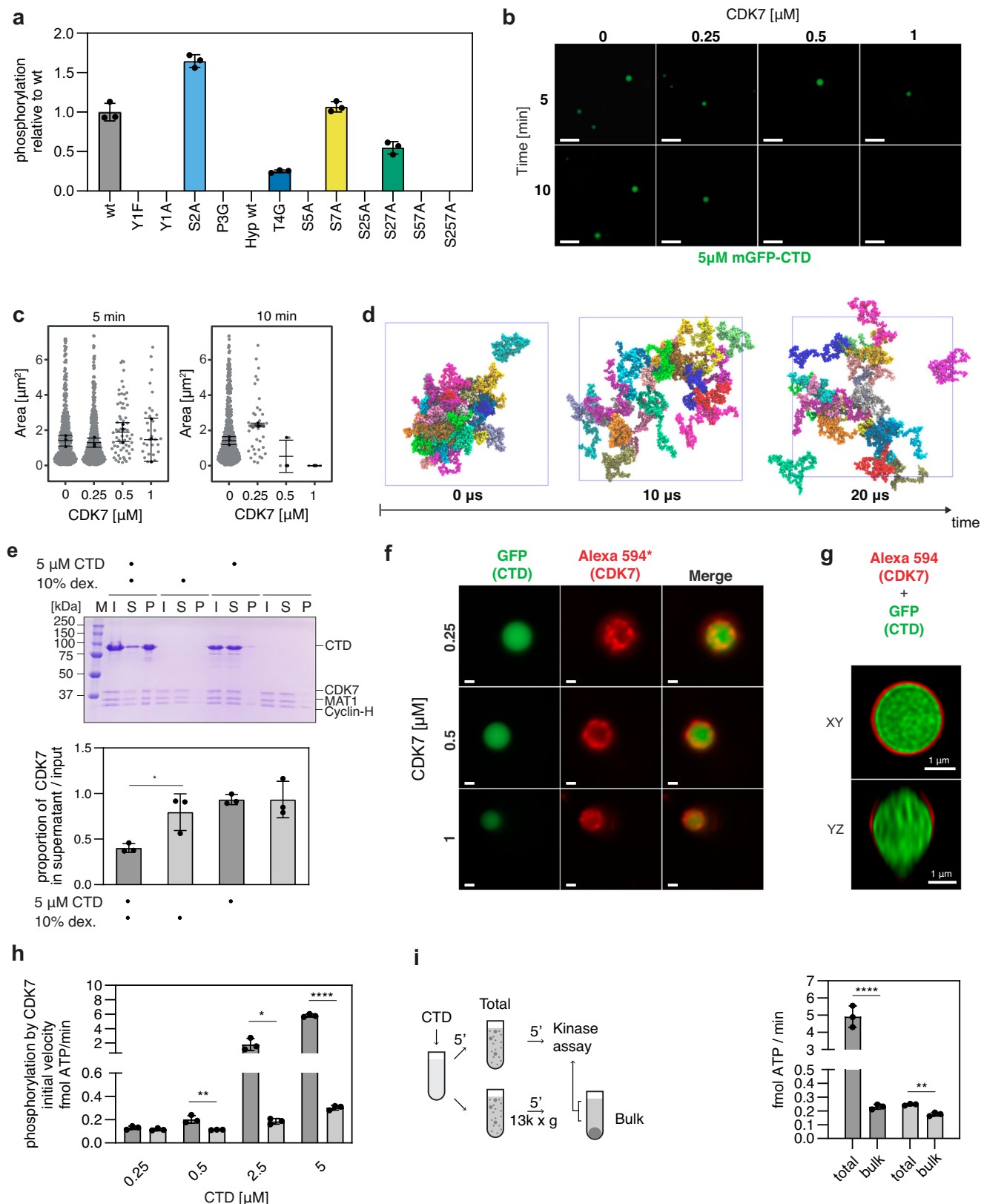

no sedimented condensates) were similar to those of reactions in the absence of dextran (Fig. 3i, Supplementary Fig. 11h, i Supplementary Fig. 12l). In contrast, the phosphorylation rate was increased 21-fold in total reaction, which contained droplets. This suggests that CTD droplets may function as condensed phase microreactors to increase local concentration of enzyme and substrate, which enhances turnover[68,69].

It is also consistent with the recent notion that kinases may be regulated by liquid-liquid phase-separation in vivo[70].

## Discussion
We generated an array of RNAPII CTD variants to reveal sequence grammar underlying CTD phase-separation. The CTD is composed of

**Fig. 3 | The impact of CTD heptad residues on phosphorylation by CDK7 and CTD phosphorylation analysis under phase-separation conditions. a** The initial reaction velocities of the human CDK7 complex with the mGFP-CTD substrate and its variants relative to the wild-type CTD. Black dots indicate the individual measurements (*n* = 3). Bars, mean; error bars, ± SD. **b** Representative micrographs of three independent in vitro LLPS kinase assays with 5 µM mGFP-CTD (green) and 0.5 mM ATP after 5 and 10 min with indicated concentrations of the CDK7 complex. Scale bar = 10 µm. **c** Quantifications of experiments shown in (**b**). Each dot represents a detected droplet, black dots indicate median of droplet size per measurement (*n* = 3). The black line and error bar represent the mean ± SD of the three medians. **d** Representative snapshots from CG condensed phase simulations of CTD molecules phosphorylated at S5. Rendering same as (**b**) in Fig. 2. **e** The CDK7 complex sedimentation assay in the presence of mGFP-CTD. A scan of representative SDS-PAGE (*top*)−(M) molecular weight marker; (I) input; (S) supernatant;

(P) pellet. Quantifications of the sedimentation assays (*bottom*) (*n* = 3), mean ± SD. **f** Representative micrographs of at least two independent in vitro LLPS assays with mGFP-CTD (green) and increasing concentration of the CDK7 complex labeled with Alexa594 (red). Merge = overlay of GFP and Alexa 594 channels. Scale bar = 1 µm. (*) The intensity of all micrographs was uniformly enhanced in Fiji[125] for better visibility. **g** Super-resolution micrographs show the CDK7 complex (red) localization at the surface of the mGFP-CTD (green) droplets. **h** Kinase assays with mGFP-CTD at different concentrations and 0.5 mM ATP with the CDK7 complex (at 5 nM), in the presence and absence of 10% dextran (*n* = 3), respectively, mean ± SD. **i** (*left*) Schematic representation of the sedimentation kinase assay workflow. **i** (*right*) Quantification of the sedimentation kinase assay (*n* = 3), mean ± SD. Statistical significance for (**e**, **h**, **i**) was determined by unpaired, two-sided *t*-test. *p*-values: (**e**) * = 0.030, (**h**) ** = 0.0093, * = 0.0292, **** ≤ 0.0001, (**i**) ** = 0.0012, **** = 0.0002. Source data are provided as a Source Data file.

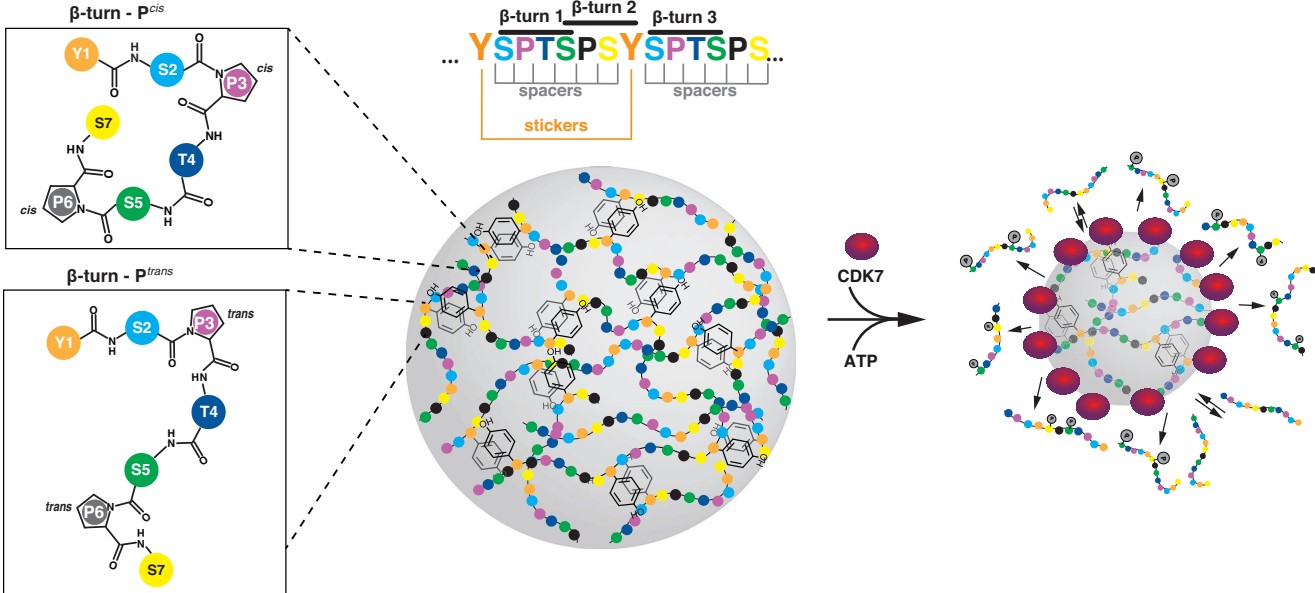

**Fig. 4 | General overview of RNAPII CTD sequence and structural features, which have influence on RNAPII CTD phase-separation.** A model of a droplet containing the CTD of RNAPII highlights the interacting tyrosine *sticker* residues separated by *spacers* (*middle*). *Spacers* can adopt distinct β-turns depending on the *cis-trans* proline isomerization (*left*). RNAPII CTD droplet dissolution upon phosphorylation by the CDK7 complex (*right*). Letters in colored circles designate amino acid residues of the CTD heptad repeat. Peptides were drawn in ChemSketch (Freeware) 2024.1.1 (Advanced Chemistry Development, Inc).

tyrosine, serine, and proline, which are enriched in intrinsically disordered proteins[71–73]. Within the framework of *sticker*-and-*spacer* model[41,74], the tyrosine residues in the CTD function as *stickers*, driving phase-separation[40,43,75–78], and they are better *stickers* than phenylalanine residues[79]. This underscores the importance of the aromatic residue at position 1 within the CTD heptapeptide motif and that the more electron-rich aromatic side-chain of tyrosine may have stronger stacking interactions[80,81]. In addition, the phenolic hydroxyl group of tyrosine can also undergo hydrogen bonding[82]. The strong interactions among tyrosine residues are attenuated by the serine residues in the *spacer* regions, acting as a buffering factor, which suppresses aggregation and maintains conditions optimal for LLPS. Additionally, our MD simulation data show that sequence variations cause the loss of local β-turn structures in the spacer regions and the decompaction of the CTD ensemble. Consequently, these CTD variants alter its phase-separation properties, as demonstrated in both experimental and MD simulation results (Figs. 1 and 2). We conjecture that the altered *cis-trans* isomerization within the hydroxyproline CTD variant modifies the propensity to form β-turns subsequently reducing the ability of CTD[Hyp] to undergo phase-separation. In line with this assertion, MD simulations showed that the *cis* conformer of both proline and hydroxyproline is crucial for driving the compactness of the CTD, whilst the extra hydroxyl group of hydroxyproline has only minor

effect on CTD compactness. Unlike proline, hydroxyproline disfavors the *cis* conformation of the prolyl-peptidyl bond and has lower energy barrier for the *cis-trans* isomerization[52]. Therefore, the different conformational properties of proline and hydroxyproline variants might be associated with the lack of co-mixing between mGFP-CTD[Hyp] and mCherry-CTD in our mixing phase-separation experiments (Fig. 1f,g). However, we note that other factors may govern co-phase-separation[83,84]. Altogether, our data suggest that conformationally restricted local structures within *spacer* regions, separating uniformly spaced tyrosine *stickers* of the CTD heptads, play crucial role in modulating interactions among these *stickers* (Fig. 4). This modulation is key to achieving a balance, which effectively facilitates LLPS. Our findings indicate that the *cis-trans* isomerization of prolines in the CTD heptapeptide motif has additional roles beyond merely controlling the timing of Ssu72-mediated dephosphorylation and the recruitment of the NNS termination complex[13,23].

Despite a generally good agreement between our in vitro and in silico experiments, there are model limitations that could account for the observed discrepancies (e.g., for the CTD[Y1F] variant). The latest reparameterization (i.e., Martini3) of the popular Martini coarse-grained model significantly mitigated the previously observed over-aggregation of soluble proteins[85]. However, when no secondary structure restraints are present (e.g., IDRs and IDPs), the global size of

the molecule is still underestimated (*i.e.*, too compact). To improve the agreement with experimental observations it has been suggested to scale down protein-protein interactions[86], especially in the presence of other biochemical moieties such as lipid membranes, or scale up protein-water interactions[87], as we did in this work. This simple solution, although in general convenient, may critically affect the aggregation behavior in a salt concentration-dependent manner[88]. Because the mapping of phenylalanine residues in Martini3 causes partial disruption of side-chain stacking interactions[89,90], it is possible that the modified balance between protein-protein and protein-water interactions led to over-aggregation driven by hydrophobic effects, in contrast with the experimentally observed decrease of the phase-separation propensity for the Y-to-F mutant[81,91,92]. A direct comparison between experimental and in silico results from atomistic simulations is complicated by the differences in the studied size and time scales. However, the atomistic simulations were essential for assessing the important effect of conformational changes in the CTD backbone related to the proline isomerization.

Our work provides a rationale for the significance of prolines within the CTD consensus heptad. We found that substituting proline with glycine at position 6, but not at position 3, impairs the solubility of the CTD. This finding strongly suggests that the proline at position 6 interferes with the formation of aggregated, potentially amyloid-like structures, as predicted by AlphaFold2 (Supplementary Fig. 1c). We hypothesize that prolines within the CTD act as a molecular switch, with their ability to sample the *cis-trans* isomerization states at the prolyl-peptidyl bond. The altering *cis* and *trans* conformations of proline in the repetitive, proline-rich CTD sequence promotes isomer diversity, preventing the formation of repetitive amyloid-like structures. This hypothesis is consistent with recent findings that pathogenic mutations leading to aggregation in the low-complexity amyloid-like kinked segments of FUS, TDP-43, hnRNPA1/A2 involve proline substitutions[93]. Similarly, introducing prolines at specific sites significantly decreases the amyloidogenic potential of human islet amyloid polypeptide[94]. Our observation also aligns with the broader concept that prolines, along with charged amino acid residues, function as anti-aggregation "gatekeepers"[95,96].

We also assessed the role of PPIases, which accelerate slow interconversion between *cis* and *trans* proline isomers[25]. Interestingly, these enzymes have been reported to exert varying effects on phase-separation: in some instances, they stimulated the process[97], whilst in others, they dissolved droplets, demonstrating an opposite effect[98]. Currently, it is unclear whether the impact of isomerases on phase-separation is driven by their binding properties or by the isomerase activity of these enzymes. In this study, we examined the effects of the single-domain PPIase enzyme PPIA and the two-domain enzyme PIN1, in which binding and catalytic functions are decoupled to a large extent[99]. The WW domain of PIN1, functioning as a specificity module, provides binding affinity for its primary phosphoserine-proline substrates but it has undetectable binding for non-phosphorylated CTD[57]. Whilst PIN1 showed no impact on CTD phase-separation, both PPIA and its catalytically inactive variant (PPIA^R55A) dissolved CTD droplets in a concentration-dependent manner. These findings collectively suggest a mechanism in which PPIA reverses phase-separation through a direct binding, independent of PPIase activity.

We showed that the CDK7 complex does not undergo phase-separation in vitro, correlating with absence of an IDR with features that would drive LLPS within its primary sequence (Supplementary Fig. 14). However, the CDK7 complex associates with the surface of CTD droplets, thereby facilitating rapid hyperphosphorylation. In contrast, the CDK9 complex operates through a different mesoscale mechanism. Its subunit, Cyclin T1, features a histidine-rich domain—an IDR, which is an integral part of hyperphosphorylated RNAPII-containing hubs in vivo, localizing the kinase inside the condensates[100]. The difference in the mechanism of action between

the CDK7 complex and the CDK9 complex may potentially be explained within the framework of the condensate-based model of transcription[34], as the kinases act at different stages of the transcription cycle. Whilst CDK7, as a subunit of the TFIIH complex, phosphorylates RNAPII during transcription initiation[4,101,102], CDK9 facilitates the transition into elongation by releasing RNAPII from promoter-proximal pausing[35,103–105]. These different mechanisms of action may therefore reflect the different chemical environments found within the promoter and gene-body molecular condensates/hubs[35,106–109].

By localizing on the surface of the CTD droplets, the CDK7 complex may leverage the distinct properties of the interface to accelerate phosphorylation by increasing substrate interaction likelihood through the law of mass action[67]. Moreover, the surface of the droplet reduces diffusion limitation for CDK7's distributive phosphorylation, potentially enabling enzyme cluster-mediated channeling to facilitate multi-site phosphorylation of the CTD[110]. This mechanism could also serve to exclude the already phosphorylated CTD stretches from the interface. Contrasting to other reports of kinase reactions inside phase-separated condensates, we did not observe a shift from distributive to a processive mode of action for CDK7 reaction with the CTD (Supplementary Fig. 12h)[111]. Additionally, it raises an interesting question about whether the arrangement of kinases on the surface of a phase-separated substrate influences phosphorylation specificity, similarly to what is observed when phosphorylation occurs inside the droplets[111].

In summary, the combination of in vitro and in silico approaches enabled us to provide insights into the role of individual amino acids within the CTD heptapeptide, highlighting the role of *cis-trans* isomerization of prolines. Moreover, we provided a framework to investigate the effect of phase-separation on the writers of the CTD code.

## Methods

### Cloning and construct design

Series of synthetic open reading frames (ORF; Invitrogen), codon-optimized for bacterial expression, encoding the wild-type human RPB1 C-terminal domain (residues 1593-1970) and its CTD variants (Supplementary Table 1) fused with a monomeric Green Fluorescent Protein (mGFP) or monomeric Red Fluorescent Protein Cherry (mCherry) at the N-terminus were cloned into 2Bc-T (Addgene #37236) expression vectors with 6xHis tag at the C-terminus by ligation-independent cloning[112]. The fragment of DNA encoding the codon-optimized CTD was subcloned from 2Bc-T into 2S-T (Addgene #29711) expression vector with a 6xHis tag and a Sumo tag at the N-terminus by ligation independent cloning. The fragment of DNA containing the ORF of human PPIA was amplified from cDNA (kindly provided by Dr. Fedor Nikulenkov, Masaryk University) by PCR and cloned into 2BT (Addgene #29666) expression vector with 6xHis tag at the N-terminus. The PPIA^R55A mutation was introduced by site-directed mutagenesis. PIN1 was a gift from Dustin Maly (Addgene #40773). The PIN1^C113S variant was generated by site-directed mutagenesis. To express the kinase module of TFIIH complex (the CDK7 complex) in insect cells, the ORFs for CDK7, MAT1, and CCNH were individually cloned into plasmid 438B (Addgene #55219), respectively. The plasmids 2Bc-T, 2S-T, 2BT, and 438B were a gift from Scott Gradia[112]. Plasmids 438B-CDK7, 438B-MAT1, and 438B-CCNH were subsequently combined using Bio-Brick Polypromoter LIC subcloning[111] into a single construct for co-expression of the three subunits of the complex from a single virus in insect cells. All constructs were verified by sequencing.

### Expression of CTD (w/o tag), mCherry-CTD, mGFP-CTD, and its variants

The corresponding expression plasmids were transformed into *E. coli* BL21-AI(DE3) cells (ThermoFisher), expressed overnight at 30 °C in TB medium. Expression was induced by 1 mM IPTG and 0.02% arabinose.

## Expression of mGFP-CTD^Hyp

Hydroxyproline mGFP-CTD was expressed in proline auxotrophic *E. coli* JM103(F⁻) cells transformed with mGFP-CTD and pTARA:500 plasmids using a modified protocol[53]. The plasmid pTara:500 was a gift from Matthew Bennett (Addgene plasmid # 60717)[113]. Starter cultures were grown at 30 °C in LB medium with 0.5% glucose, ampicillin, and chloramphenicol. M9 medium (1X M9 salts, 100 μg/ml all amino acids, 2 mM MgSO$_4$, 0.1 mM CaCl$_2$, 0.5% glucose) was inoculated with the starter culture and grown until OD$_{600}$ = 1 at 30 °C. Cultures were washed three times in M9 medium lacking proline and incubated for 90 min at 30 °C to deplete proline in the cells. Cultures were induced by the addition of 0.02% arabinose and 1 mM IPTG, osmotically shocked by the addition of 600 mM NaCl and provided with 40 mM trans-4-hydroxy-L-proline (Carbosynth). After 16 h of expression, cells were harvested by centrifugation.

## Expression of PIN1 and PIN1^C113S

The PIN1 expression plasmids were transformed into *E. coli* BL21 CodonPlus (DE3) -RIPL cells (Agilent Technologies) and expressed in 2X TY medium at 37 °C. After reaching OD = 0.6, the cultures were induced by the addition of 1 mM IPTG, and expression was carried out at 16 °C for 18 h.

## Expression of PPIA wt and PPIA^R55A

The PPIA expression plasmids were transformed into *E. coli* BL21(DE3) cells (ThermoFisher) and grown in LB medium with ampicillin at 37 °C. After reaching OD = 0.6, the cultures were induced by the addition of 1 mM IPTG, and expression was carried out at 16 °C for 16 h.

## Expression of the CDK7 complex

To generate viruses enabling the production of proteins in insect cells, the coding sequences and the necessary regulatory sequences of the constructs were transposed into bacmid using *E. coli* strain DH10bac. The viral particles were obtained by transfection of the bacmids into the Sf9 cells using FuGENE Transfection Reagent and further amplification. The CDK7 complex was expressed in 300 mL of High Five insect cells (infected at $1 \times 10^6$ cells/ml) with the corresponding P1 virus at a multiplicity of infection >1. The cells were harvested 48 h post-infection, washed with 1x PBS, and stored at −80 °C.

## Protein purification

**Purification of CTD (w/o tag), mCherry-CTD, mGFP-CTD, and its variants.** Cells obtained from a two-liter culture were resuspended in lysis buffer (50 mM Tris pH 8, 500 mM NaCl, 1 mM DTT, 10 mM imidazole, 3 mg/ml lysozyme, cOmplete™ Mini EDTA-free Protease Inhibitor Cocktail (Roche)) and lysed by sonication. The lysate was clarified by centrifugation. The supernatant was incubated with Ni-NTA agarose beads. Ni-NTA purification was performed on a gravity flow column, and protein sample was eluted with elution buffer (50 mM Tris pH 8, 500 mM NaCl, 1 mM DTT, 400 mM imidazole). Fractions containing CTD were pooled together, mixed with TEV protease (1 mg/ml), and dialyzed against the dialysis buffer (50 mM Tris pH 8, 300 mM NaCl, 1 mM DTT, 1 mM PMSF) overnight. The dialyzed sample was incubated with Ni-NTA Agarose and reverse Ni-NTA affinity purification was performed on a gravity flow column by collecting the flow through and washing the beads with dialysis buffer supplemented with 5-, 10-, and 500-mM imidazole, respectively. Size exclusion chromatography was performed on the Superdex 200 column (Cytiva) previously equilibrated with SEC buffer (25 mM HEPES-OH pH 7.4, 220 mM NaCl, 0.5 mM TCEP).

**Purification of PIN1, PIN1^C113S, PPIA, PPIA^R55A.** Cells obtained from 2 L culture were resuspended in lysis buffer (50 mM Tris pH 8, 500 mM NaCl, 1 mM DTT, 10 mM imidazole, 3 mg/ml lysozyme, cOmplete™ Mini EDTA-free Protease Inhibitor Cocktail (Roche)) and lysed by

sonication. The lysate was clarified by centrifugation. The supernatant was incubated with Ni-NTA agarose beads. Ni-NTA purification was performed on a gravity flow column, and protein sample was eluted with elution buffer (50 mM Tris pH 8, 500 mM NaCl, 1 mM DTT, 50/100/150/200/300 mM imidazole gradient). Fractions containing PIN1 proteins were pooled together, mixed with TEV protease (1 mg/ml), and dialyzed against the dialysis buffer (50 mM Tris pH 8, 300 mM NaCl, 1 mM DTT, 1 mM PMSF) overnight. The dialyzed sample was incubated with Ni-NTA Agarose and reverse Ni-NTA affinity purification was performed on a gravity flow column by collecting the flow through and washing the beads with dialysis buffer supplemented with 5-, 10-, and 500-mM imidazole, respectively. Size exclusion chromatography was performed on the Superdex 75 column (Cytiva) previously equilibrated with SEC buffer (25 mM HEPES-OH pH 7.4, 220 mM NaCl, 0.5 mM TCEP).

**Purification of the CDK7 complex.** Pellets of Hi5 insect cells were resuspended in ice-cold lysis buffer (50 mM Tris pH 8.0; 500 mM NaCl; 0.4% Triton X-100; 10% (w/v) glycerol; 10 mM imidazole; 1 mM DTT), containing protease inhibitors (0.66 μg/ml pepstatin, 5 μg/ml benzamidine, 4.75 μg/ml leupeptin, 2 μg/ml aprotinin, and 25 U benzonase per ml of lysate). The resuspended cells were gently shaken for 10 min at 4 °C. To aid the lysis, cells were briefly sonicated. The cleared lysate was passed through 2 mL of Ni-NTA beads (Qiagen), equilibrated with buffer (50 mM Tris-Cl, pH 8, 500 mM NaCl, 10 mM imidazole, and 1 mM DTT). Proteins were eluted with an elution buffer (50 mM Tris-Cl, pH 8, 500 mM NaCl; 1 mM DTT, and 400 mM imidazole). The elution fractions containing proteins were pooled, mixed with TEV protease, and dialyzed against dialysis buffer (50 mM Tris-Cl, pH 8; 500 mM NaCl; 1 mM PMSF, and 1 mM DTT). A second affinity step was used to remove the TEV protease and other contaminants, the flowthrough protein was pooled and concentrated. Size exclusion chromatography was performed on Superose 6 column with SEC buffer (25 mM Tris-Cl pH7.5; 220 mM NaCl, 1 mM DTT). Fractions containing pure complex were pooled and concentrated. Purified protein that was used in phase-separation studies was flash-frozen and stored at −80 °C. CDK7 that was not to be used in phase-separation assays was supplemented with 10% (w/v) glycerol before flash-freezing and storage.

**Mass spectrometry.** The identity of purified mCherry-CTD, mGFP-CTD, and its variants was confirmed via mass spectrometry (For more details see Supplementary Methods). All raw data are available via the PRIDE partner repository[114] with the dataset identifier PXD049700.

**Turbidity assay.** mGFP-CTD and CTD (w/o tag) purified in the phase-separation reaction buffer (25 mM HEPES-OH 7.4, 220 mM NaCl, 0.5 mM TCEP) were added to 10% (w/v) Dextran T500 (Pharmacosmos)) up to final concentrations between 0.612 μM and 10 μM. Turbidity was measured on SpectraMax iD5 microplate reader at 600 nm in 96-well plate with flat bottom. Turbidity assay was done in biological triplicates (independent in vitro LLPS assays).

**Fluorescence microscopy (FM).** For all following fluorescence microscopy experiments the imaging was performed on a Zeiss Axio Observer.Z1 inverted microscope and 64x objective with water immersion. All assays were done in biological triplicates (independent in vitro LLPS assays).

**FM—CTD variants.** Purified mGFP-CTD variants were added to the pre-prepared phase-separation reaction mixture (25 mM HEPES-OH 7.4, 220 mM NaCl, 0.5 mM TCEP, and 10% (w/v) Dextran T500 (or 10% PEG-8000 (AppliChem)) to a final concentration of 2.5 μM, 5 μM, and 10 μM, respectively. The mixture was vortexed for 5 s. One μl of the mixture was put on a glass microscope slide and imaged within five minutes. Four micrographs per variant and condition were used for

further analysis. Dissolution of the droplets was tested by addition of hexane-1,6-diol (Sigma-Aldrich) up to 10% concentration to the LLPS reaction and incubation for 30 min.

**FM—mGFP-CTD and CTD (w/o tag).** Purified mGFP-CTD and CTD (w/o tag) were mixed in ratios of 1:0, 1:10, and 100, respectively, and added to the pre-prepared phase-separation reaction mixture (25 mM HEPES-OH 7.4, 220 mM NaCl, 0.5 mM TCEP, and 10% (w/v) Dextran T500) up to a final concentration of 10 μM. The mixture was vortexed for 5 s. One μl of the mixture was put on a glass microscope slide and imaged within 5 min. Four micrographs per variant and condition were used for further analysis.

**FM—mixing experiments (mGFP-CTD, mCherry-CTD, mGFP-CTD$^{Hyp}$).** Purified mCherry-CTD was added to the pre-prepared phase-separation reaction mixture (25 mM HEPES-OH 7.4, 220 mM NaCl, 0.5 mM TCEP, and 10% (w/v) Dextran T500) to a final concentration of 5 μM. The phase-separation was started by vortexing the mixture for 5 s. In the following step, mGFP-CTD and mGFP-CTD$^{Hyp}$, respectively, were added to the final concentration of 5 μM (the added volume did not exceed 20 % of the volume of the whole reaction). One μl of the mixture was put on a glass microscope slide and imaged within five minutes. This step was repeated after 10 and 20 min, respectively. Four micrographs per construct and condition were used for analysis.

**FM—mixing experiments (mGFP-CTD, PIN1, PPIA).** Purified mGFP-CTD was added to the pre-prepared phase-separation reaction mixture (25 mM HEPES-OH 7.4, 220 mM NaCl, 0.5 mM TCEP, and 10% (w/v) Dextran T500) up to a final concentration of 5 μM. The phase-separation was started by vortexing the mixture for 5 s. In the following step, PIN1, PIN1$^{C113S}$, PPIA, or PPIA$^{R55A}$ was added up to the final concentration of 80 μM or 160 μM (the added volume did not exceed 20 % of the volume of the whole reaction). After 5 minutes of incubation, 1 μl of the mixture was put on a glass microscope slide and imaged. This step was repeated after 15 min (total incubation time 20 min). Five micrographs per construct and condition were used for further analysis.

**FM—mixing experiments (mGFP-CTD, the CDK7 complex).** Purified mGFP-CTD was added to the pre-prepared phase-separation reaction mixture (25 mM HEPES-OH 7.4, 220 mM NaCl, 0.5 mM TCEP, and 10% (w/v) Dextran T500) to a final concentration of 5 μM. The phase-separation was started by vortexing the mixture for 5 s. In the following step, the CDK7 complex was added up to final concentrations of 0.25 μM, 0.5 μM, and 1 μM, respectively (the added volume did not exceed 20 % of the volume of the whole reaction). Alternatively, the CDK7 was added up to final concentrations of 0.25 μM, 0.5 μM, and 1 μM, respectively, to the reaction mixture before initiating the phase-separation. One μl of the mixture was put on a glass microscope slide and imaged after five minutes of incubation. Five micrographs per construct and condition were used for further analysis.

**FM—kinase assays.** The phase-separation reaction mixture (25 mM HEPES-OH 7.4, 220 mM NaCl, 0.5 mM TCEP, and 10% (w/v) Dextran T500) was supplemented with 0.5 mM ATP and mGFP-CTD was added to a final concentration of 5 μM. The phase-separation was started by vortexing the mixture for 5 s. In the following step, the CDK7 complex was added up to final concentrations of 0.25 μM, 0.5 μM, and 1 μM respectively. One μl of the mixture was put on a glass microscope slide and imaged within five minutes. The rest of the phase-separation reaction mixture was incubated for an additional 10 min at room temperature and 1 μl of the mixture was put on a glass microscope slide and imaged within five minutes. In parallel, 5 μl of the reactions were inactivated at the indicated time points by mixing with SDS

Loading Dye and denaturing at 95 °C for 5 min, followed by SDS PAGE analysis. Five micrographs per construct and condition were used for further analysis.

**FM—image analysis.** Condensate properties were analyzed using CellProfiler 4.2.1. software[115]. Initial objects (droplets) identification for analysis was done based on diameter (4-70 px/0.413 -7.5 μm), and pixel intensity (median of pixel intensity values (0.1–1)). Otsu's method[116] was used for image thresholding. Picked objects were further filtered based on shape (eccentricity (filter 0.6)—ratio between main axis length and the foci of the ellipse, where 0 = circle, and 1 = line segment) and intensity (median intensity values of pixels (0.1–1)). Calculated parameters for filtered objects: Area, object count per picture, object count per measurement, or integrated intensities (sum of pixel signal in the detected droplet) were obtained by analyzing the GFP channel (mCherry and GFP channels for mixing experiments with mGFP-CTD$^{Hyp}$). R[117] and R-studio[118] with tidyverse[119] package were used to process the data obtained from Cell-profiler analysis. The values for droplet areas were converted from the px to μm based on the metadata of the micrographs (0.010645 μm² = 1px). Graphs for the figures were plotted using ggplot2[120], ggbeeswarm[121], ungeviz[122], and ggpubr[123] packages. Statistical analysis for the mean intensity and droplet count comparison (derived from medians from three measurements) was done using unpaired, two-sided $t$-test[124]. Symbols * indicating statistical significance follow the convention: ns: $p > 0.05$, *: $p \leq 0.05$, **: $p \leq 0.01$, ***: $p \leq 0.001$, ****: $p \leq 0.0001$. Indicated in the figures are only symbols for $p$-values *: $p \leq 0.05$ and smaller. Graphs were plotted for the y-axis range between 0 and 7.5 for the Area parameter. Representative micrographs for the figures were generated in Fiji[125], the size of the presented views is 512 × 512 pixels—colocalization experiments with the mGFP-CTD and the CDK7 complex 100 × 100 pixels (the size of all source data images is 2048 × 2048 pixels). Where stated, the intensity of the micrographs was uniformly enhanced for the purpose of presentation. Raw fluorescence microscopy data (czi files) used for analyses and generation of the representative micrographs (images picked from those used for analysis) are deposited on ZENODO[126].

**The CDK7 complex labeling with Alexa 594.** The purified CDK7 complex was labeled with Alexa Fluor® 594 Conjugation Kit/Alexa Fluor® 594 Labeling Kit (ab269822) according to the manufacturer's manual.

**The CDK7 complex colocalization with the mGFP-CTD droplets.** Purified mGFP-CTD was added to the pre-prepared phase-separation reaction mixture (25 mM HEPES-OH, pH 7.4, 220 mM NaCl, 0.5 mM TCEP, and 10% (w/v) Dextran T500) up to a final concentration of 5 μM. The phase-separation was started by vortexing the mixture for 5 s. In the following step, CDK7 complex (at ratio 1:10 labeled to non-labeled) was added up to the final concentration 0.25 μM, 0.5 μM, and 1 μM (the added volume did not exceed 20 % of the volume of the whole reaction). Alternatively, the CDK7 complex (at ratio 1:10 labeled to non-labeled) was added up to final concentrations of 0.25 μM, 0.5 μM, and 1 μM, respectively, to the reaction mixture before starting the phase-separation. One μl of the mixture was put on a glass microscope slide and imaged after five minutes of incubation. The imaging was performed on a Zeiss Axio Observer.Z1 inverted microscope and 64x objective with water immersion. Representative micrographs were generated in Fiji[125].

For the super-resolution microscopy, 1 μl of the mixture was spotted onto a glass microscope slide and imaged with the Elyra 7 inverted microscope with lattice SIM and 64x objective with oil immersion. Images were acquired and processed in ZEN Software (Black Edition) using SIM$^1$ processing (fixed medium, fixed strong)[127]. Representative images were generated in the Imaris Software (v. 10.0, Bitplane, Oxford Instruments, Abingdon-on-Thames, UK).

### In vitro kinase assays with γ-[$^{32}$P]-ATP

The kinase assays with the CTD variants were performed in triplicates in 10 μl reactions containing the mGFP-CTD variant (all at 1 μM) as the substrate and the CDK7 complex (at 1 nM) in a reaction buffer containing 40 mM Tris pH 7.5, 20 mM MgCl$_2$, 50 mM NaCl, 1 mM DTT, and 0.1 mg/ml BSA. The kinase was added from a 10-fold concentrated stock in kinase dilution buffer consisting of 25 mM Tris pH 7.5, 150 mM NaCl, 10% (w/v) glycerol, 0.01% NP-40, 0.1 mg/ml BSA, and 1 mM DTT. The reactions were initiated by the addition of 1 μl of 5 mM ATP containing 1 μCi γ-[$^{32}$P]-ATP (Hartmann Analytik) and incubated at 23 °C for 5 min. The reactions were stopped with the addition of ½ volume of the SDS Loading Dye and incubated at 90 °C for 3 min. The phosphorylated proteins were separated from unincorporated ATP by SDS-PAGE, the gels were exposed onto a phosphor screen overnight and the screens were scanned using Amersham Typhoon Biomolecular Imager (Cytiva). The signal was quantified with ImageQuant (Fujifilm), and the signal of the CTD variant proteins was normalized to the signal of the wild-type CTD substrate to maintain uniformity between experiments. The results were plotted in GraphPad Prism 9.3.1.

The Kinase assays for phase-separation were performed with varying concentrations of mGFP-CTD wild-type substrate and fixed concentration of 5 nM the CDK7 complex. Kinase dilution buffer was the same as for the variant phosphorylation assays with the omission of glycerol, kinase reaction buffer consisted of 25 mM HEPES-OH 7.4, 220 mM NaCl, 20 mM MgCl$_2$, and 0.5 mM TCEP. The kinase reaction and phase-separation were simultaneously initiated by the addition of dextran-ATP mixture or ATP and water, for total concentration of 10% (w/v) Dextran T500 and 0.5 mM ATP and 0.1 μCi/μl γ-[$^{32}$P]-ATP. Reactions were incubated at 23 °C for 5 min and inactivated and analyzed in the same way as described above. Reactions were performed in biological triplicates. Unpaired t-tests were performed in GraphPad Prism 9.3.1. Significance is reported for p-values *: $p \leq 0.05$ and smaller.

### Sedimentation LLPS assays

Phase-separation assays with CDK7 and the mGFP-CTD were performed in triplicates and set up in 50 μl volume in a buffer containing 25 mM HEPES-OH 7.4, 220 mM NaCl, 20 mM MgCl$_2$, and 0.5 mM TCEP. The CDK7 complex was used at 0.5 μM concentration. Phase-separation was initiated by the addition of the mGFP-CTD (at 5 μM) and 10% (w/v) dextran T500 (final concentration). Reactions were briefly vortexed and incubated at 23 °C for 5 min. The input fraction was analyzed by SDS-PAGE. Reactions were centrifuged at 13,000 × g for 5 min at 23 °C to pellet the droplets. After centrifugation, the supernatant was removed for SDS-PAGE analysis and the pellet was dissolved in an equal volume of SDS-PAGE Loading Dye. The SDS-PAGE samples were separated on a 15% SDS-PAGE gels, followed by Coomassie staining. Reactions were performed in biological triplicates. The intensity of bands corresponding to the subunits of the CDK7 complex in the individual fractions was measured by GelAnalyzer (GelAnalyzer 23.1.1; www.gelanalyzer.com), summed, and plotted in GraphPad Prism 9.3.1. When CTD was analyzed in isolation, the reactions were performed as described above, and the intensity of the CTD band was measured by GelAnalyzer and plotted using GraphPad Prism 9.3.1. Unpaired t-tests were performed in GraphPad Prism 9.3.1. Significance is reported for p-values *: $p \leq 0.05$ and smaller.

### Sedimentation kinase assay with γ-[$^{32}$P]-ATP

To compare the contributions of the droplet phase and bulk phase in the kinase reactions in phase-separation conditions, a sedimentation reaction was performed as described in[66]. Briefly, the kinase reactions were set up in 15 μl volume in the presence or absence of 10% (w/v) Dextran T500, 0.5 mM ATP, and 0.1 μCi/μl γ-[$^{32}$P]-ATP was included in the master mix. After the addition of 5 μM CTD (final concentration), the reactions were briefly vortexed and incubated at 23 °C for 5 min. A subset of the reactions was centrifuged at 13,000 × g for 5 min to separate the pelleted droplet phase from the bulk supernatant phase. The supernatant from the centrifuged reactions was collected and inactivated. A fraction of the reactions not subjected to centrifugation was also inactivated at the same timepoint. The samples were separated from unincorporated ATP by SDS-PAGE and the signal was quantified as described above. Reactions were performed in biological triplicates. Note that the reactions containing 10% (w/v) dextran were diluted 10-fold prior to loading onto SDS-PAGE to avoid saturation of the signal. Data was plotted and unpaired t-tests were performed in GraphPad Prism 9.3.1. Significance is reported for p-values *: $p \leq 0.05$ and smaller.

### Computational simulations

Molecular Dynamics (MD) simulations of various CTD single mutants were conducted at both atomistic and coarse-grained (CG) resolution. To capture the effect of CDK7 kinase activity, we also included CTD constructs phosphorylated at either S5 or S7. In addition, atomistic simulations were used to address the effect of isomerization of the prolyl peptide bond by imposing cis or trans isomerization states to (hydroxy)prolines at position 3 and 6, individually and combined (see Supplementary Table 3).

All constructs featured a short PS sequence at the start and uncharged N- and C- termini. In all-atom simulations, these termini were respectively capped with acetyl (ACE) and N-methyl amide (NME) groups. Modifications like amino acid substitutions, proline isomerization states, and capped termini were implemented using PyMol[128] and tleap from AmberTools[129]. All simulations were performed using the Gromacs 2021 simulation package[130] and the plumed 2.7 library[131].

### Coarse-grained models

CG simulations used the Martini3 model[85], with the ε parameter of the Lennard-Jones (LJ) potential between protein and water beads increased by a factor of 1.1. Such enhanced protein solvation was shown to better reproduce the global dimensions of various IDPs[87]. Martini3 lacks parametrization for phosphorylated amino acids. Therefore, we approximated the side chain of phosphoserine with a charged bead of type Q5 (−1e) or D (−1.5e). The backbone-side chain bond length was increased from 0.287 to 0.319 nm to mimic the all-atom structure of phosphorylated serine.

CTD structures, comprising 52 repeats of the consensus heptapeptide, were generated using the Martinize2 script[132] as random coils (i.e., no secondary structure assignment). We conducted two types of simulations: 1) single chain simulations aimed at evaluating the structural properties of the isolated chain, which serve as predictors of phase-separation propensity[133]; and 2) condensed phase simulations focused on directly studying phase-separation of CTD molecules.

In single chain simulations, a CTD construct was placed in a 42.5 × 42.5 × 42.5 nm$^3$ box with around 620,000 water beads, 150 mM NaCl, and counterions for electroneutrality. As for condensed phase simulations, 10, 20, 50, or 80 equilibrated copies of the non-phosphorylated domains were randomly inserted in a 45 × 45 × 45 nm$^3$ box, with 600,000 to 720,000 water beads and 150 mM NaCl, yielding protein concentrations of 200 to 1800 μM. For selected protein concentrations and CTD variants, including the phosphorylated domains, we also simulated systems starting from preformed condensates, the configuration of which was taken from the end of 20 μs long simulations of CTD$^{cons}$ (see Supplementary Table 3).

All CG simulations used the New-RF simulations parameters[134], wherein the Coulomb interactions were treated with the reaction field method and infinite dielectric constant beyond the cutoff of 1.1 nm. At the same cut-off distance, the LJ potential was shifted to zero. Apart from the equilibration where the Berendsen barostat[135] was used, temperature and pressure control mirrored all-atom simulations (see below), although with coupling time of 1 and 12 ps, respectively. Each final production run extended for 20 μs, employing a time step of

either 20 or 25 fs, depending on the stability of the system. Condensed phase simulations were run in triplicates.

## All-atom models

To address the intrinsic limitations of the used CG model and explore processes like hydrogen bonding, secondary structure formation, and prolyl isomerization, we supplemented our simulations with higher-resolution all-atom models.

Building upon recent work reporting significantly improved structural properties of IDPs[136], our atomistic simulations employed the ff99SB Amber force field[137] in combination with OPC water, a 4-site model calibrated for precise representation of water molecules electrostatics[138]. Models for doubly deprotonated phosphorylated amino acids (i.e., charge −2$e$) and hydroxyproline were taken from Amber-compatible parametrizations[139,140].

To capture both short and long-range inter- and intramolecular interactions, while keeping a reasonable size of the simulated system, two di-heptad CTD constructs were randomly inserted in an $8 \times 8 \times 8$ nm$^3$ cubic box. Each system was then solvated with approximately 16,000 water molecules and 150 mM NaCl, ensuring charge neutrality with additional sodium counterions when necessary. Energy minimization involved 5000 steps of steepest descent method, initially with and then without positional restraints on protein-heavy atoms, using a force constant of 1000 kJ mol$^{-1}$ nm$^{-2}$ and a maximum force convergence criterion of 500 kJ mol$^{-1}$ nm$^{-1}$. Next, 9 ns of stepwise equilibration in the NPT ensemble progressively relaxed the CTD molecules by applying the positional restraints on all protein-heavy, backbone, C$\alpha$, and no atoms. This was followed by 4 μs of unrestrained NPT MD.

The temperature of the protein and solvent was kept constant at 300 K using two separate velocity-rescaling thermostats[141] with time constants of 0.5 ps (0.1 ps during equilibration). A pressure of 1 bar was isotropically maintained via the Parrinello-Rahman barostat[142] and a coupling time of 2 ps. The LJ potential was shifted to zero at 0.9 nm, with long-range dispersion corrections for energy and pressure. The same cutoff of 0.9 nm was used for short-range electrostatic interactions, while long-range electrostatics was treated with the particle mesh Ewald (PME) method[143] and grid spacing of 0.12 nm in the reciprocal space. Protein covalent bonds and water molecules were constrained using LINCS[144] algorithms, allowing an integration time step of 2 fs. All simulations were run in triplicates to minimize the sensitivity to the initial conditions. Configurations were saved every 20 ps.

## Data analysis

Trajectory manipulation, data visualization, and routine data analysis were performed using Gromacs 2021[130], MDAnalysis 2[145], and VMD[146]. Unless stated otherwise, only the final 3 and 10 μs of the all-atom and CG simulations were considered for analysis, and with the exception of single chain CG simulations, the results are presented as mean ± standard deviation (SD) over three replicas and all CTD constructs. The details of more specialized analyses are described in the following. *CG Single Chain Simulations*. To quantitatively evaluate the structural behavior of single-chain proteins, we used the Flory scaling exponent ($v$). This quantity can be derived from the relation[147]: $r_{ij} = b|i\text{-}j|^v$ where $r_{ij}$ is the average distance between residue i and j, $|i\text{-}j|$ represents the residues separation in peptide bonds, and $b$ is the Kuhn length, which is set at 0.55 nm[133]. $v$ values exceeding 0.5 indicate an extended conformation, while lower values suggest compactness. For reference, a fully expanded coil state corresponds to $v = 3/5$[148], while a globular state is denoted by $v = 1/3$[149].

To calculate mean and SD of the Flory exponent we considered only residue pairs separated by more than 200 peptide bonds. *CG Condensed Phase Simulations*. We tracked the formation of droplet-like aggregates by monitoring the size of the protein clusters over time. A CTD molecule was considered to be part of a cluster when it had more than a predefined threshold number of contacts with any other cluster member. In particular, a contact was defined as any pair of protein beads within the distance cutoff of 0.6 nm. Three distinct threshold values were used: 20, 50, and 80. *Atomistic Simulations*. The presence of β-turns was assessed using the simple geometric criterion of the distance between C$\alpha$ atoms of the residues at positions 1 and 4 being less than 0.7 nm[150]. As for the analysis of the intramolecular hydrogen bonding network, we employed the HydrogenBondAnalysis[151] utility from MDAnalysis.

The following MD simulations data are deposited in Zenodo [https://doi.org/10.5281/zenodo.10696484][126]: a) input files for all simulated systems, b) raw data (e.g., time series), c) processed data (e.g., distributions) and structural coordinates displayed in figures, and d) analysis scripts.

## Reporting summary

Further information on research design is available in the Nature Portfolio Reporting Summary linked to this article.

## Data availability

All the processed data are presented in the main article and the Supplementary Information. Source data are provided in the Source data file. The raw fluorescence microscopy data (used for the analyses and representative micrographs) are deposited on ZENODO[126]. The mass spectrometry proteomics data are deposited to the ProteomeXchange Consortium via the PRIDE partner repository[114] with the dataset identifier PXD049700. Plasmids used in this study are available upon request from the corresponding author (richard.stefl@ceitec.muni.cz). The simulation input, output, and parameter files are available from Robert Vacha (robert.vacha@muni.cz) due to their large file size. The simulation input files are available on ZENODO[126]. Source data are provided with this paper.

## Code availability

There is no original code used in this study. All MD simulations input files, raw data for figures, and analysis scripts are deposited on ZENODO[126].

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

## Acknowledgements

We thank Dr. Milan Ešner and Dr. Jakub Pospíšil (CELLIM of CEITEC, Masaryk University) for their assistance with the imaging experiments and Dr. Wojciech Jesionek (CELLIM of CEITEC, Masaryk University) for his assistance with image analysis. We thank Dr. Ondrej Šedo (Proteomics Core Facility, CEITEC, Masaryk University) for his assistance with mass spectrometry. We thank Dr. David Kovář (Lochsmidt Laboratories, Masaryk University) for the assistance with the turbidity assays. We acknowledge funding from the Czech Science Foundation to R.S. (grant no. 21-24460S) and to M.S. (grant no. 20-21581Y), and the European Research Council to R.S. (grant no. 649030) and to R.V. (grant no. 101001470). We acknowledge support by the MEYS CR to R.S. (grant no. CZ.02.01.01/00/22_008/0004575 RNA for therapy), the project National Institute of Virology and Bacteriology to R.V. (Programme EXCELES, LX22NPO5103)—Funded by the European Union—Next Generation EU. We acknowledge the core facility CELLIM supported by MEYS CR (LM2023050 Czech-BioImaging). We acknowledge CEITEC Proteomics Core Facility of CIISB, Instruct-CZ Centre, supported by MEYS CR (LM2023042, e-INFRA CZ (ID:90254)). Computational resources were provided by the CESNET, CERIT Scientific Cloud, and IT4 Innovations National Supercomputing Center by MEYS CR through the e-INFRA CZ (ID:90254).

## Author contributions

Conceptualization: M.S., R.V., R.S., Methodology: K.L., M.M., F.L.F., M.S., Investigation: K.L., M.M., F.L.F., M.S., Visualization: K.L., M.M., F.L.F., R.S., Funding acquisition: M.S., R.V., R.S., Project administration: R.V., R.S., Supervision: M.S., R.V., R.S., Writing—original draft: K.L., M.M., F.L.F., R.S., Writing—review & editing: K.L., M.M., F.L.F., M.S., R.V., R.S.

## Competing interests

The authors declare no competing interests.
