## [Transparent Peer Review file · Nature Communications]

Sequence and Structural Determinants of RNAPII CTD Phase-separation and Phosphorylation by CDK7

Corresponding Author: Professor Richard Stefl

Version 0:

Reviewer comments:

Reviewer #1

(Remarks to the Author)

This is an interesting article that uses carefully controlled biochemical assays to uncover new insights about the molecular grammar underlying the pol II CTD, which implies functional roles in the regulation of transcription via phase separation. In my opinion, such studies are needed given the many uncertainties that result from interpretation of cell-based data. Many interesting results were reported, and the investigators generated an impressive set of reagents to biochemically test which CTD residues are important for phase separation and/or phosphorylation by CDK7. Key insights included the importance of the spacing of Y1 "stickers", prerequisite S5 phosphorylation for S7 phosphorylation by CDK7, and the importance of prolines in the CTD sequence. One limitation is that the assays presented here are done only in vitro and there is no cell-based work. However, cell-based work cannot directly determine the contributions of interacting partners or other molecules, and expression of select CTD mutants would be problematic and cause unintended consequences in cells. Here, the assays can be completely controlled therefore allowing direct conclusions, which is an advantage. I have the following concerns and comments for the authors to consider in a revision.

Main points:

1. The potential importance of CH-pi interactions was not made clear, nor can I convince myself that these interactions will make a meaningful contribution to CTD phase separation. The experiments that begin to address this issue are incomplete (e.g. the P3G mutant will not eliminate all potential CH-pi interactions) and therefore I suggest to modify this section with further explanation or additional data, or remove entirely. I don't consider addressing this issue a key deliverable for this article.
2. It is extremely challenging to rigorously address proline isomerization, and I commend the authors for the insights that they have made here with the hydroxyproline derivatives. However, it is not mentioned that the introduction of hydroxyl groups onto each proline will introduce new potential for electrostatic and H-bonding interactions that could be disruptive compared to WT CTD. On line 183-183 the authors conclude that "cis-trans isomerization of prolines are key for CTD phase separation" but this seems incompletely verified, in my opinion. The authors should back off such a strong statement as it remains an open question. I want to emphasize that this criticism does not significantly detract from the results, or from my enthusiasm for the experimental design and the overall impact of the findings.
3. mGFP is 238 residues in length while the 52 repeat heptad CTD is ~ 360 residues. The mGFP-CTD fusion is nearly ~ 40% structured. The authors state that the tags "do not affect the LLPS properties of the CTD" but it should be noted that the tags may shift Csat. Did the authors try or consider tagging with fluorescent molecules instead of a large protein domain?
4. It is interesting that for the Y1A mutant, authors tested up to 45 uM to prove that, even at high concentrations, it does not phase separate. Yet for the Hyp mutant, the authors stopped at 10 uM (which can be near the Csat for some proteins) and concluded it did not phase separate. One additional higher condition could be helpful to thoroughly demonstrate a "complete loss" of the ability to phase separate.
5. An estimation of Csat for the CTD would be helpful. Related to this point, the authors utilize hexanediol to disrupt droplets but this compound has rightly fallen out of favor as a means to assess phase separation. In our opinion a more rigorous

measure is to confirm that droplet size increases with increasing [CTD], which the authors already have data to support (e.g. Fig 1d). Instead of hexanediol, the authors could also show a response by modifying salt, pH, or temperature.

Other comments:

- The consensus YSPTSPS sequence of the CTD was used for the MD simulations. It would be interesting and more biologically relevant to include the native human CTD sequence in the simulations.
- The articles Guo/Young Nature 2019 543 and Boehning/Zweckstetter NSMB 2018 833 should be cited in the section about CDK7 reversing phase separation via phosphorylation.
- The experiments evaluating if CTD phosphorylation by CDK7 is accelerated under phase separation conditions was correlative. That's ok, but perhaps the authors could change the title of this section to read "Phase-separation appears to accelerate CTD phosphorylation"
- Line 399: Edit to read "...phosphorylates RNAPII during transcription initiation..."

Reviewer #2

(Remarks to the Author)

Linhartová et al. present a comprehensive study aiming to dissect how the amino acid composition of the tandem repeats within RNA Polymerase II's C-terminal domain regulates phase separation in vitro. The authors purify numerous mutants of the C-terminal domain formed by 52 repeats to perform in vitro phase separation assays, conduct in silico studies to support their findings, and explore the role of known enzymatic regulators, Cyclin-dependent kinase 7 and two isomerases. The biological question is highly relevant – how does multivalency spatiotemporally regulate transcription? The dual approach of in vitro and in silico studies is appealing and, as far as I can tell, without an alternative, to understand the self-assembly properties under the concept of protein phase separation in high detail. Nice conclusions are drawn regarding the importance of prolines, the regulation of self-assembly by phosphorylation, and the observation that phase separation of the target accelerates the phosphorylation process itself.

Some general comments:

1. In my perspective, the integrated approach of in vitro assays paired with simulations is very exciting with the potential to create novel insights. Fundamentally, simulations alone always provide a model – whether this model is right or wrong is challenging to prove. Since the authors provide a very systematic approach in vitro, I think it would be of high interest to many readers, beyond Polymerase II, and also maximize transparency, to make a more systematic side-by-side comparison between the in vitro results and the simulations. In simple terms, have the authors received simulation results that did not match the self-assembly phenomena observed in vitro? A transparent overview would be helpful and nurture future developments, interesting for discussion.
2. The idea that proline isomerization is a regulator of phase separation could be an exciting principle. The conclusions are not so clear, as the experimental system generally reaches its boundaries. For example, the authors find that hydroxy groups in the repeat change the phase separation behavior (e.g., Mutants S2A, S5A), but the proline derivative in the experiments carries an additional hydroxy group beside the shifted isomer confirmation. What causes the effects? The hydroxy group or the isomer? More systematic simulations, not only with the isomer but also the proline carrying the hydroxy group, could strengthen the point. Reading the paper, I got the impression that the authors would like to make stronger statements about the isomers but decided to focus on phosphorylation because the evidence is stronger. Intuitively, I would say that the paper is fine with or without the isomer story, which currently feels half-cooked, despite the tremendous experimental work, and could either be strengthened and stay as part of the main manuscript or moved to extended data (e.g., Figure 1B, which otherwise sets the wrong context).
3. The imaging procedure of droplets gains value due to the massively parallel approach and relative comparisons between the mutants. The authors provide the necessary evidence to talk about phase separation, including the reversibility and the formation of droplets that often show a higher density and can be pelleted. To be precise, the manuscript includes a gel of the WT CTD showing that the induction of phase separation drives the protein into the pellet fraction, validating the imaging experiments that quantify droplet formation. The authors also make the point that the interaction strength of the repeat is a double-edged sword and must be tuned to prevent amyloid-like assemblies. In the method section, the authors write: "Picked objects were further filtered based on shape (eccentricity) and intensity (median intensity)". Given that the phase separation assays are a central element in this work, I would appreciate more detail here. How much of the total fluorescent intensity was filtered out this way? Does this create a bias between liquid-like droplets (round), aggregates (fuzzy), and amyloid-like assemblies (fuzzy)? The point is to make the manuscript more transparent. More systematic measures about shape and size should be provided to give the readers a clearer idea of how much protein phase-separated, stayed in the soluble fraction, or aggregated. Since the authors classify the fluorescent signal, this could be done on the image analysis level or, if easier, by pelleting to estimate the critical point of phase separation for the most important mutants. The images alone of a few droplets carry very little information, and the description of the classification is too vague.
4. The kinase assays are beautiful and make a strong point about physiological relevance.
5. In my experience, in vitro phase separation assays include many changing elements that include the hardening of droplets, droplet growth, etc. My question is, does the onion-like condensate structure in Figures 3g and 3f change if the order of events is changed, or if the droplets get less and more time to form? Can the authors provide field of views with more examples?

Reviewer #3

(Remarks to the Author)

See attached file.

In this work, Linhartova et.al investigate the sequence-determinants of phase separation of the C-terminal domain of RNA Polymerase II. The authors postulate that the presence of aromatic residues, as well as the cis-trans isomerization of Pro residues within the repeat unit sequence of RNAPII are critical determinants of its phase separation, with other residues, Ser and Thr playing a role in modulating the dynamics. The authors further investigate phosphorylation of RNAPII by Cyclin-dependent kinase 9 (CDK9) and report higher phosphorylation in the phase separated state implying the importance of the drivers of LLPS, Tyr and Pro, in phosphorylation as well. The experimental observations are supported by molecular dynamics (MD) simulations at coarse-grained (CG) and atomistic resolution which provide insight into the stabilization and destabilization of condensates as well as secondary structure effects upon mutation/phosphorylation. Overall, the study is well done and provides insight into how isomerization of Pro can alter phase behavior. Further, the findings relating to the action of CDK9 as well as the isomerases are quite interesting and would be of broad interest to the field. However, while the experiments as well as simulations are well conducted and comprehensive, we have some comments regarding the interpretation of the data and are not entirely convinced by some of the conclusions drawn from them. Below are the major comments/concerns.

1. In the abstract (line 29-30), the authors say,
“Serine residues within the CTD consensus sequence influence droplet morphology and inhibit aggregation.”

We feel that while the article extensively investigates phase separation, the insights provided into aggregation are not sufficient to justify the inclusion of this line within the abstract and suggest the authors focus more on the phase behavior aspect with the findings related to aggregation primarily in the discussion section.

2. In the introduction (line 75-78) and the discussion (lines 339-340), the authors say,
“The highly repetitive nature and uniform patterning of the tyrosine residues in the CTD sequence align with the sticker-and-spacer model used in phase-separation of IDPs and multidomain proteins.”
and
“In line with the previous observations, we found that the uniformly spaced aromatic stickers are required for the CTD phase-separation”

We believe that to make an argument for the uniform patterning being important for phase separation, the authors must demonstrate that non-uniform patterning influences the phase behavior through the generation of a shuffled variant that breaks the uniformity. Alternatively, the authors could omit the argument that the uniform patterning of Tyr residues is important and merely state that RNAPIII can be considered using the sticker-spacer model.

3. In the results section (line 195-196), the authors hypothesize that the Ser residue at position 7 has a negligible change in phase separation as compared to wild-type as this residue is the least conserved. We do not believe that this serves as an adequate justification and encourage the authors to provide an alternate hypothesis using physics-based rationalization as to why they may observe position dependence of Ser mutations (e.g. contacts from simulations)

4. In the results section, the authors investigate the role of Pro in the phase separation of RNAPIII. The authors postulate two main contributors about Pro, protein-protein interactions, and the effect of Pro isomers on conformations. While the data clearly show that Pro is important, we believe the conclusion that the effect is due to isomers is not supported or is not easy to follow when considering the results. Our reasoning is as follows:

- The authors perform Pro to Gly mutations and observe reduced droplet formation leading to the conclusion that protein-protein interactions are not primarily responsible. However, Pro and Gly are quite different in their properties and hence interactions with other residues. Further, while both residues are believed to inhibit the formation of secondary structure, Gly due to the absence of the sidechain could have vastly different effects on chain conformations as compared to Pro. We believe that this is not a 1:1 comparison and does not rule out the effects of protein-protein interactions involving Pro in determining phase behavior.
- The authors proceed to compare the partitioning of WT and Hyp variants into WT droplets and see that Hyp does not partition concluding that conformations are responsible for this difference. We are not convinced that this is purely dependent on chain conformation and could purely be due to the lower homotypic interactions of Hyp (evidenced by its lack of phase separation) leading to less tendency to partition within condensates of WT which would be in line with recent work on the co-phase separation of IDRs (<https://doi.org/10.1038/s41557-023-01423-7>, <https://doi.org/10.1038/s41557-024-01456-6>).
- The final section deals with the action of isomerases. The authors show how rather than catalytic activity, the direct interaction of isomerases with the proteins is responsible for the dissolution of condensates. We would expect that if the cis-trans isomerization is an important determinant of

LLPS, the catalytically active isomerase would more avidly dissolve condensates by favoring the trans isomer in WT (like the Hyp variant which doesn't undergo LLPS), but the authors find the opposite.

Overall, the combination of the above makes it hard to follow the main conclusions one should draw from the data and would benefit from more clarification regarding expectations vs. observations or a rewrite with milder language (e.g. saying that the data shows isomerization is important rather than saying isomerization is more important than interactions)

5. In the results section, the simulation data seems to show certain trends that we find interesting and encourage the authors to comment on.

- In the case of the aromatic variants, the authors observe that Y to F mutations in the CG model lead to more dense droplets and more contacts as compared to Y (at higher concentrations). The experimental data shows a reduction in phase separation with Y to F, but the simulations are the opposite (saturation concentration for the F mutant would be lower than Y). The Y to F mutation has been studied quite extensively with numerous single-bead per residue models adequately capturing the trend (<https://doi.org/10.1002/pro.4094>, <https://doi.org/10.1073/pnas.2111696118>, [10.1038/s43588-021-00155-3](https://doi.org/10.1038/s43588-021-00155-3)). We ask the authors to clarify whether the current CG model adequately captures this well-known trend. Further, we do not fully understand why the relative contacts between Y and F would be concentration dependent in the MD simulation.

- In the investigations of the different Ser and Thr mutants on secondary structure, we find it interesting that the effects often seem to be on beta sheets not involving the mutated residue (sheet 2 for S2A for example) and ask that the authors comment/discuss why this may be.

6. In the results section, when discussing the phosphorylation, the S2A mutant leads to an increase of almost 1.5 times. This seems to be the only mutant for which an increase is observed, however, it is currently not discussed in the manuscript. We ask that the authors discuss this in the results section and provide some insight into why this may be happening and potential implications.

7. In the results as well as the discussion section, the authors argue that the solvation of Ser in the "spacer" regions coupled with strong interactions among tyrosine residues together dictate aggregation vs. LLPS. We point the authors to the following experimental data (<https://doi.org/10.1021/bi00507a030>) that demonstrates the solvation of Tyr is more favorable than Ser. Therefore, this does not serve as an explanation for the differences between Tyr and Ser. We encourage the authors to rethink this model and propose or justify why favorable (but less than Tyr) solvation of Ser attenuates aggregation.

Aside from the above, we have the following minor corrections to suggest.

1. Fig1C, the S7A mutant is incorrectly labeled as S5A
2. In the results section, line 132, the authors state that their investigation reveals that the tags do not affect phase separation. We suggest using "minimally effect" rather than "do not".
3. In the discussion, where the authors discuss the importance of Tyr in phase separation (line 341-344), we would like to point to authors to the following work highlighting the role of Tyr ([10.1038/s43588-021-00155-3](https://doi.org/10.1038/s43588-021-00155-3)) as well the hydrogen bonding of Tyr with other amino acids in LLPS (<https://doi.org/10.1073/pnas.2000223117>).

Reviewer #4

(Remarks to the Author)

Version 1:

Reviewer comments:

Reviewer #1

(Remarks to the Author)

I thank the authors for their responses. I have no further comments or concerns.

Reviewer #2

(Remarks to the Author)

I believe the authors have added the necessary simulations and analysis, and have rephrased where needed. I have no further requests.

Reviewer #3

(Remarks to the Author)

The authors have done a wonderful job of addressing concerns from the previous round. I have not hesitation in recommending publication of the paper in the current form.

Reviewer #4

(Remarks to the Author)

We thank the reviewers for their time and valuable feedback, which has helped us to improve the presentation of our results and better underscore their significance. Below, we provide our responses to the specific issues raised by the reviewers, along with the summary of changes made to the manuscript. The reviewers' comments are presented in black. Our responses and the summary of changes are colored in blue.

REVIEWER COMMENTS

Reviewer #1 (Remarks to the Author):

This is an interesting article that uses carefully controlled biochemical assays to uncover new insights about the molecular grammar underlying the pol II CTD, which implies functional roles in the regulation of transcription via phase separation. In my opinion, such studies are needed given the many uncertainties that result from interpretation of cell-based data. Many interesting results were reported, and the investigators generated an impressive set of reagents to biochemically test which CTD residues are important for phase separation and/or phosphorylation by CDK7. Key insights included the importance of the spacing of Y1 “stickers”, prerequisite S5 phosphorylation for S7 phosphorylation by CDK7, and the importance of prolines in the CTD sequence. One limitation is that the assays presented here are done only in vitro and there is no cell-based work. However, cell-based work cannot directly determine the contributions of interacting partners or other molecules, and expression of select CTD mutants would be problematic and cause unintended consequences in cells. Here, the assays can be completely controlled therefore allowing direct conclusions, which is an advantage. I have the following concerns and comments for the authors to consider in a revision.

We thank the reviewer for their encouraging and constructive assessment of our work. The point-by-point responses to the comments are provided below.

Main points:

1. The potential importance of CH-pi interactions was not made clear, nor can I convince myself that these interactions will make a meaningful contribution to CTD phase separation. The experiments that begin to address this issue are incomplete (e.g. the P3G mutant will not eliminate all potential CH-pi interactions) and therefore I suggest to modify this section with further explanation or additional data, or remove entirely. I don't consider addressing this issue a key deliverable for this article.

We have followed the reviewer's suggestion and removed the discussion of CH-pi interactions in the context of the P3G variant in the revised manuscript.

2. It is extremely challenging to rigorously address proline isomerization, and I commend the authors for the insights that they have made here with the hydroxyproline derivatives. However, it is not mentioned that the introduction of hydroxyl groups onto each proline will introduce new potential for electrostatic and H-bonding interactions that could be disruptive compared to WT CTD. On line 183-183 the authors conclude that "cis-trans isomerization of prolines are key for CTD phase separation" but this seems incompletely verified, in my opinion. The authors should back off such a strong statement as it remains an open question. I

want to emphasize that this criticism does not significantly detract from the results, or from my enthusiasm for the experimental design and the overall impact of the findings.

We thank the reviewer for pointing out this comment and their kind words regarding our attempt to address the role of proline isomerization with the hydroxyproline derivatives. Although hydroxyproline is a close proline derivative with a shifted population of isomers, we agree with the reviewer that proline-to-hydroxyproline substitutions introduce an additional hydroxyl group that could potentially alter protein-protein interactions. This point is also mentioned by Reviewer 2, who suggested addressing this by additional comparative simulations, as there are no experimental means for resolving this question. To gain additional insights into what causes the effects – “The hydroxyl group or the isomer?” – we have conducted additional simulations, which are summarized in the new **Supplementary Figure 6**. The simulations with proline- and hydroxyproline-containing CTD showed that the additional hydroxyl groups do not disrupt interactions but, instead, create additional ones. However, these interactions have only a minor effect on the CTD compactness, which is primarily determined by the *cis* isomerization state.

Whilst these additional simulations support our conclusion regarding the importance of *cis-trans* isomerization for CTD phase-separation, we have followed the reviewers’ suggestions by softening our statement and explicitly addressing the role of the hydroxyl group of CTD^{Hyp}.

Changes made to the manuscript:

(page 9)

Altogether, our experiments suggest that the aromaticity of the Y1 residue and possibly the *cis-trans* isomerization of prolines are key for CTD phase-separation.

(page 11/12)

To test the critical role of *cis-trans* proline isomerization on LLPS, we performed additional all-atom MD simulations of CTD^{cons} and its hydroxyproline variant (CTD^{consHYP}). Specifically, we explored combinations of the isomerization state of prolines and hydroxyprolines at positions 3 and 6 of the heptad repeat: *trans-trans* (i.e. major state), *cis-trans*, *trans-cis*, and *cis-cis* (note that only the first and last combinations were included for CTD^{consHYP}). Overall, we observed that all *cis* isomers showed greater compactness (**Fig. 2c and Supplementary Figs. 6a and 7a**), a smaller radius of gyration (**Fig. 2d and Supplementary Figs. 6b and 7b**), and a general increase of the per-residue intramolecular interaction energies across the entire CTD di-heptad (**Fig. 2e and Supplementary Figs. 6c and 7d**). This behavior was particularly observed in the case of the double *cis* isomers (CTD^{cisP3,6} and CTD^{cisHYP3,6}). Interestingly, both CTD^{consHYP} and CTD^{cisHYP3,6} variants showed minor shifts of conformational populations towards more compact structures when compared with the proline counterparts. This shift likely originates from the additional local interactions involving the extra OH group of hydroxyproline (**Supplementary Figs. 7d and 8a**). However, the shift from the *cis* isomer is significantly larger.

3. mGFP is 238 residues in length while the 52 repeat heptad CTD is ~ 360 residues. The mGFP-CTD fusion is nearly ~ 40% structured. The authors state that the tags "do not affect the LLPS properties of the CTD" but it should be noted that the tags may shift Csat. Did the authors try or consider tagging with fluorescent molecules instead of a large protein domain?

We thank the reviewer for their insightful comment regarding the possible impact of the mGFP tag on the LLPS properties of the CTD. To explore this further, we purified tag-free CTD and compared its LLPS properties with those of mGFP-CTD. We conducted three different experiments: (i) droplet assays with tag-free CTD monitored by differential interference contrast (DIC) microscopy, (ii) mixing droplet assays with labeled and unlabeled CTD monitored by microscopy with statistical analyses, and (iii) turbidity assays of tagged and tag-free CTD, respectively, via OD₆₀₀ measurements. Altogether, the new data, which are included in **Supplementary Fig. 1**, show that the differences in the LLPS properties between tagged and non-tagged CTD are negligible.

Changes made to the manuscript:

(page 7)

We also tested whether tagging of CTD with mGFP and mCherry, respectively, affects CTD's ability to undergo LLPS and whether the tags themselves can phase-separate under conditions used in the assays. We found that the tags alone did not phase-separate (**Supplementary Fig. 1e**) and that the differences in LLPS properties between the tagged and non-tagged CTD were negligible. (**Supplementary Fig. 1f-i**).

4. It is interesting that for the Y1A mutant, authors tested up to 45 μM to prove that, even at high concentrations, it does not phase separate. Yet for the Hyp mutant, the authors stopped at 10 μM (which can be near the C_{sat} for some proteins) and concluded it did not phase separate. One additional higher condition could be helpful to thoroughly demonstrate a "complete loss" of the ability to phase separate.

We thank the reviewer for pointing this out. For the hydroxyproline variant, we originally tested the LLPS properties within the range of concentrations used for all variants as stated in the original manuscript. We did not, however, attempt to concentrate the mGFP-CTD^{Hyp} as the purified sample contains a byproduct with low hydroxyproline content, as determined by mass-spectrometry analysis, which could not be separated from the mGFP-CTD^{Hyp} (**Supplementary Fig. 1b**; commented in the section "Preparation of RNAPII CTD variants").

In the revised manuscript, we tested mGFP-CTD^{Hyp} at concentrations of 15 and 40 μM , respectively. At these concentrations we did observe droplet formation, which may be ascribed to the presence of the byproduct with low hydroxyproline content. The new data are included in **Supplementary Fig. 2d**. At these concentrations of mGFP-CTD^{Hyp}, the byproduct may reach C_{sat} , which might be similar to the one of mGFP-CTD.

Changes made to the manuscript:

(page 8)

However, the mGFP-CTD^{Hyp} variant did phase-separate at concentrations of 15 μM and higher. (**Supplementary Fig. 2d**). The phase-separation observed at higher mGFP-CTD^{Hyp} concentrations may be due to the presence of a byproduct with low hydroxyproline content in our sample (**Supplementary Fig. 1b**), which could still undergo phase-separation.

5. An estimation of C_{sat} for the CTD would be helpful. Related to this point, the authors utilize hexanediol to disrupt droplets but this compound has rightly fallen out of favor as a means to assess phase separation. In our opinion a more rigorous measure is to confirm that droplet size increases with increasing [CTD], which the authors already have data to support

(e.g. Fig 1d). Instead of hexanediol, the authors could also show a response by modifying salt, pH, or temperature.

We thank to the reviewer for the comment. We would like to point the reviewer to a recent publication from the Zweckstetter lab by Flores-Solis, et al. Nat Commun, 2023 in which the estimation of Csat for the CTD and the impact of pH, salt, and temperature on LLPS properties was reported. Therefore, we focused our efforts on other properties of RNAPII CTD phase-separation.

Other comments:

- The consensus YSPTSPS sequence of the CTD was used for the MD simulations. It would be interesting and more biologically relevant to include the native human CTD sequence in the simulations.

In our coarse-grained molecular dynamics simulations, we used the consensus YSPTSPS sequence repeated 52 times as a model for the native human CTD sequence. This choice facilitated direct comparisons with all-atom molecular dynamics simulations that utilized a smaller number of consensus heptad repeats. It is important to highlight that only position 7 of the heptad sequence shows poor conservation in the second half of the human CTD (repeats 25-52).

- The articles Guo/Young Nature 2019 543 and Boehning/Zweckstetter NSMB 2018 833 should be cited in the section about CDK7 reversing phase-separation via phosphorylation.

We have cited the two papers in the section about CDK7 reversing phase-separation via phosphorylation as suggested by the reviewer.

- The experiments evaluating if CTD phosphorylation by CDK7 is accelerated under conditions was correlative. That's ok, but perhaps the authors could change the title of this section to read "Phase-separation appears to accelerate CTD phosphorylation"

We have changed the title of this section as suggested by the Reviewer.

- Line 399: Edit to read "...phosphorylates RNAPII during transcription initiation..."

We have edited the text as suggested by the Reviewer.

Reviewer #2 (Remarks to the Author):

Linhartová et al. present a comprehensive study aiming to dissect how the amino acid composition of the tandem repeats within RNA Polymerase II's C-terminal domain regulates *in vitro*. The authors purify numerous mutants of the C-terminal domain formed by 52 repeats to perform *in vitro* assays, conduct *in silico* studies to support their findings, and explore the role of known enzymatic regulators, Cyclin-dependent kinase 7 and two isomerases. The biological question is highly relevant – how does multivalency spatiotemporally regulate transcription? The dual approach of *in vitro* and *in silico* studies is appealing and, as far as I can tell, without an alternative, to understand the self-assembly properties under the concept of protein in high detail. Nice conclusions are drawn regarding the importance of prolines, the regulation of self-assembly by phosphorylation, and the observation that of the target accelerates the phosphorylation process itself.

Some general comments:

1. In my perspective, the integrated approach of *in vitro* assays paired with simulations is very exciting with the potential to create novel insights. Fundamentally, simulations alone always provide a model – whether this model is right or wrong is challenging to prove. Since the authors provide a very systematic approach *in vitro*, I think it would be of high interest to many readers, beyond Polymerase II, and also maximize transparency, to make a more systematic side-by-side comparison between the *in vitro* results and the simulations. In simple terms, have the authors received simulation results that did not match the self-assembly phenomena observed *in vitro*? A transparent overview would be helpful and nurture future developments, interesting for discussion.

We thank the reviewer for the recommendation and agree that a critical comparison of the results would be beneficial for the audience. Therefore, the following text has been added to the Discussion section of the revised manuscript:

Changes made to the manuscript:

Despite a generally good agreement between our *in vitro* and *in silico* experiments, there are model limitations that could account for the observed discrepancies (e.g., for the CTD^{Y1F} variant). The latest reparameterization (*i.e.*, Martini3) of the popular Martini coarse-grained model significantly mitigated the previously observed over aggregation of soluble proteins¹²³. However, when no secondary structure restraints are present (*e.g.*, IDRs and IDPs), the global size of the molecule is still underestimated (*i.e.*, too compact). To improve the agreement with experimental observations it has been suggested to scale down protein-protein interactions⁸⁶, especially in the presence of other biochemical moieties such as lipid membranes, or scale up protein-water interactions¹²⁴, as we did in this work. This simple solution, although in general convenient, may critically affect the aggregation behavior in a salt concentration dependent manner.⁸⁷ Because the new mapping of phenylalanine residues in Martini3 causes partial disruption of side-chain stacking interactions^{88,89}, it is possible that the modified balance between protein-protein and protein-water interactions led to over aggregation driven by hydrophobic effects, in contrast with the experimentally observed decrease of the phase-separation propensity for the Y-to-F mutant.^{82,90,91} A direct comparison between experimental and *in silico* results from atomistic simulations is complicated by the differences in the studied size and time scales. However, the atomistic simulations were essential for assessing the important effect of conformational changes in the CTD backbone related to the proline isomerization.

2. The idea that proline isomerization is a regulator of phase separation could be an exciting principle. The conclusions are not so clear, as the experimental system generally reaches its boundaries. For example, the authors find that hydroxy groups in the repeat change the phase separation behavior (e.g., Mutants S2A, S5A), but the proline derivative in the experiments carries an additional hydroxy group beside the shifted isomer confirmation. What causes the effects? The hydroxy group or the isomer? More systematic simulations, not only with the isomer but also the proline carrying the hydroxy group, could strengthen the point. Reading the paper, I got the impression that the authors would like to make stronger statements about the isomers but decided to focus on phosphorylation because the evidence is stronger. Intuitively, I would say that the paper is fine with or without the isomer story, which currently feels half-cooked, despite the tremendous experimental work, and could either be strengthened and stay as part of the main manuscript or moved to extended data (e.g., Figure 1B, which otherwise sets the wrong context).

We agree with the reviewer that additional simulations could further strengthen our suggestion that proline isomerization could be an important regulator of phase-separation. To this end, we have performed additional all-atom simulations of the hydroxyproline-containing CTD in both *cis* and *trans* isomerization states. The results of these simulations are included in **Supplementary Figs. 6 and 7** and are discussed in the manuscript as outlined in our response to **Reviewer 1, point 2**.

In summary, we observed that, although the additional interactions from the extra hydroxyl group of hydroxyprolines locally increase the structural compactness, the effect of *cis-trans* isomerization on the conformational population is significantly larger. This is in line with our hypothesis that proline isomerization plays an important role in CTD phase-separation.

3. The imaging procedure of droplets gains value due to the massively parallel approach and relative comparisons between the mutants. The authors provide the necessary evidence to talk about phase separation, including the reversibility and the formation of droplets that often show a higher density and can be pelleted. To be precise, the manuscript includes a gel of the WT CTD showing that the induction of phase separation drives the protein into the pellet fraction, validating the imaging experiments that quantify droplet formation. The authors also make the point that the interaction strength of the repeat is a double-edged sword and must be tuned to prevent amyloid-like assemblies. In the method section, the authors write: “Picked objects were further filtered based on shape (eccentricity) and intensity (median intensity)”. Given that the phase separation assays are a central element in this work, I would appreciate more detail here. How much of the total fluorescent intensity was filtered out this way? Does this create a bias between liquid-like droplets (round), aggregates (fuzzy), and amyloid-like assemblies (fuzzy)? The point is to make the manuscript more transparent. More systematic measures about shape and size should be provided to give the readers a clearer idea of how much protein phase-separated, stayed in the soluble fraction, or aggregated. Since the authors classify the fluorescent signal, this could be done on the image analysis level or, if easier, by pelleting to estimate the critical point of phase separation for the most important mutants. The images alone of a few droplets carry very little information, and the description of the classification is too vague.

We thank the reviewer for suggesting how to improve the presentation of the data. We provided additional details for the image analysis description in the Methods section (*FM – image analysis*).

To demonstrate the impact of the eccentricity filters used in this work, we re-examined the same images for the mGFP-CTD and its variants without applying any eccentricity filter. As a result, only two criteria were used to select droplets: (i) diameter between 4-70px (0.413 - 7.5 μm), (ii) fluorescence intensity higher than the calculated background. For mGFP-CTD and most of its variants, we observed no significant difference in the data analysis with and without eccentricity filter, detecting a similar number of droplets regardless of its use (see figure below). However, for the mGFP-CTD^{S2,5,7A} and mGFP-CTD^{S2,5A} variants, we observed differences in the number of detected objects (droplets) between the filtered and non-filtered datasets (see summarizing Figure below and **Supplementary Fig. 4g**). *We note that the two variants were reported to form non-spherical aggregates in the original version of our manuscript.* Comparing the number of droplets in the final analysis with and without the eccentricity filter may provide an estimate of how many droplets in our dataset are round and regularly shaped, versus those that may represent the aging effect, and thus were discarded during the analysis.

Changes made in the Supplementary Fig. 4g: graphs summarizing average droplet count for the analysis of mGFP-CTD and mGFP-CTD^{S2,5,7A} with and without the eccentricity filter.

Changes made in the manuscript: (page 10)

We note that these aggregates were discarded during our image analysis due to their shape (**Supplementary Fig. 4g**).

Comparison of the average droplet count for the (a) mGFP-CTD, mGFP-CTD^{P3G}, mGFP-CTD^{Hyp}, (b) mGFP-CTD, mGFP-CTD^{Y1F}, mGFP-CTD^{Y1A}, (c) mGFP-CTD, mGFP-CTD^{S2A}, mGFP-CTD^{S5A}, and mGFP-CTD^{S7A}, (d) mGFP-CTD, mGFP-CTD^{T4G} and (e) mGFP-CTD, mGFP-CTD^{S2,5A}, mGFP-CTD^{S2,7A}, and mGFP-CTD^{S5,7A} with and without (nf) the eccentricity filter at three concentrations. The black dots indicate the sum of the droplets that were detected by the analysis per measurement ($n=3$).

We appreciate the suggestion to include the pelleting approach to estimate the critical point of phase-separation for mutants. However, this approach would be insensitive to distinguishing droplets from aggregates. Rather than assessing the intensity of the fluorescence signal, we believe that evaluating the shape of the droplets is a more reliable indicator whether they are round or fuzzy.

4. The kinase assays are beautiful and make a strong point about physiological relevance.

We appreciate the reviewer's positive feedback on the kinase assays.

5. In my experience, *in vitro* assays include many changing elements that include the hardening of droplets, droplet growth, etc. My question is, does the onion-like condensate structure in Figures 3g and 3f change if the order of events is changed, or if the droplets get less and more time to form? Can the authors provide field of views with more examples?

Following the reviewer's suggestion, we have performed additional experiments in which we preincubated mGFP-CTD and the non-labeled CDK7 complex prior to phase-separation. We found only small differences in droplet size, with the droplets being smaller (**Supplementary Fig. 12c,d**). The change of events resulted in the same localization of the CDK7 complex at the surface of CTD droplets (**Supplementary Fig. 11e**).

We enclosed here a larger field of views with more examples, as requested (see below). The images presented here are 512x512 pixels in size, whilst the source data are 2048x2048 pixels (all source data are deposited in ZENODO (DOI: 10.5281/zenodo.10696484)). The images in the manuscript, **Fig. 3g** and **Supplementary Fig. 11d**, are 100x100 pixels in size.

Changes made in the manuscript:

(page 15)

“We observed only small differences when we induced the mGFP-CTD phase-separation before adding the CDK7 complex (**Supplementary Fig. 12 a,b**), compared to the scenario in which we mixed the CDK7 complex with mGFP-CTD before inducing phase-separation. In the latter experiment, smaller droplets formed, and this effect was more pronounced with increasing concentration of CDK7 (**Supplementary Fig. 12 c,d**).“

(page 16)

“The CDK7 complex remained localized at the surface of the droplets also when it was preincubated with mGFP-CTD before inducing the phase-separation (**Supplementary Fig. 11e**).”

The LLPS assay with mGFP-CTD and increasing concentrations of the CDK7 complex, labeled with Alexa594 (1:10) with mixing of mGFP-CTD and the CDK7 complex before the start of the phase-separation (*) the intensity of all fluorescence micrographs was uniformly enhanced in Fiji for better visibility (*left*) and after the start of the mGFP-CTD phase-separation (*right*). White circles highlight the droplets presented in **Fig. 3f** and **Supplementary Fig. 11d, e**.

Reviewer #3 (Remarks to the Author):

In this work, Linhartova et.al investigate the sequence-determinants of phase separation of the C-terminal domain of RNA Polymerase II. The authors postulate that the presence of aromatic residues, as well as the cis-trans isomerization of Pro residues within the repeat unit sequence of RNAPII are critical determinants of its phase separation, with other residues, Ser and Thr playing a role in modulating the dynamics. The authors further investigate phosphorylation of RNAPII by Cyclin-dependent kinase 9 (CDK9) and report higher phosphorylation in the phase separated state implying the importance of the drivers of LLPS, Tyr and Pro, in phosphorylation as well. The experimental observations are supported by molecular dynamics (MD) simulations at coarse-grained (CG) and atomistic resolution which provide insight into the stabilization and destabilization of condensates as well as secondary structure effects upon mutation/phosphorylation. Overall, the study is well done and provides insight into how isomerization of Pro can alter phase behavior. Further, the findings relating to the action of CDK9 as well as the isomerases are quite interesting and would be of broad interest to the field. However, while the experiments as well as simulations are well conducted and comprehensive, we have some comments regarding the interpretation of the data and are not entirely convinced by some of the conclusions drawn from them. Below are the major comments/concerns.

1. In the abstract (line 29-30), the authors say, “Serine residues within the CTD consensus sequence influence droplet morphology and inhibit aggregation.”

We feel that while the article extensively investigates phase separation, the insights provided into aggregation are not sufficient to justify the inclusion of this line within the abstract and suggest the authors focus more on the phase behavior aspect with the findings related to aggregation primarily in the discussion section.

We have followed the Reviewer’s suggestion and removed this sentence from the abstract.

2. In the introduction (line 75-78) and the discussion (lines 339-340), the authors say, “The highly repetitive nature and uniform patterning of the tyrosine residues in the CTD sequence align with the sticker-and-spacer model used in phase-separation of IDPs and multidomain proteins.” And “In line with the previous observations, we found that the uniformly spaced aromatic stickers are required for the CTD phase-separation”

We believe that to make an argument for the uniform patterning being important for phase separation, the authors must demonstrate that non-uniform patterning influences the phase behavior through the generation of a shuffled variant that breaks the uniformity. Alternatively, the authors could omit the argument that the uniform patterning of Tyr residues is important and merely state that RNAPolIII can be considered using the sticker-spacer model.

We have followed the Reviewer’s suggestion and removed the discussion on the uniform patterning of Y1.

3. In the results section (line 195-196), the authors hypothesize that the Ser residue at position 7 has a negligible change in phase separation as compared to wild-type as this residue is the least conserved. We do not believe that this serves as an adequate justification and encourage the authors to provide an alternate hypothesis using physics-based rationalization as to why they may observe position dependence of Ser mutations (e.g. contacts from simulations)

The reviewer is right that there are more possible reasons for different behavior of Ser variants and, therefore, we removed our speculation on the origin of the weak influence of Ser7 substitution.

4. In the results section, the authors investigate the role of Pro in the of RNAPoIII. The authors postulate two main contributors about Pro, protein-protein interactions, and the effect of Pro isomers on conformations. While the data clearly show that Pro is important, we believe the conclusion that the effect is due to isomers is not supported or is not easy to follow when considering the results. Our reasoning is as follows:

- The authors perform Pro to Gly mutations and observe reduced droplet formation leading to the conclusion that protein-protein interactions are not primarily responsible. However, Pro and Gly are quite different in their properties and hence interactions with other residues. Further, while both residues are believed to inhibit the formation of secondary structure, Gly due to the absence of the sidechain could have vastly different effects on chain conformations as compared to Pro. We believe that this is not a 1:1 comparison and does not rule out the effects of protein-protein interactions involving Pro in determining phase behavior.

We agree with the reviewer and have omitted the interpretation for the P3G variant in the revised manuscript (this was also suggested by the **Reviewer 1**).

- The authors proceed to compare the partitioning of WT and Hyp variants into WT droplets and see that Hyp does not partition concluding that conformations are responsible for this difference. We are not convinced that this is purely dependent on chain conformation and could purely be due to the lower homotypic interactions of Hyp (evidenced by its lack of) leading to less tendency to partition within condensates of WT which would be in line with recent work on the co-phase separation of IDRs (<https://doi.org/10.1038/s41557-023-01423-7>, <https://doi.org/10.1038/s41557-024-01456-6>).

The reviewer is correct to emphasize that co-phase-separation, and in particular the organization of the resulting condensates, strongly depends on the complex interplay between homo- and heterotypic interactions. To gain further insights into what makes CTD^{Hyp} less prone to phase-separation (the hydroxyl group or the isomer conformations), we performed additional atomistic simulations of CTD di-heptads with Pro-to-Hyp substitutions (also suggested by Reviewer 2), which showed that the additional intramolecular interactions from the extra -OH group of CTD^{Hyp} perturb the conformational populations of both *cis* and *trans* isomerization variants (see new **Supplementary Fig. 7**). It is therefore plausible that also the intermolecular interactions (or multivalency) of Hyp differ from proline and, consequently, its phase-separation behavior. However, MD simulations showed that the *cis* conformer of both proline and hydroxyproline is crucial for driving the overall CTD conformations, whilst the extra hydroxyl group of hydroxyproline has only minor effect on it. To conclude, we agree that the different chemical nature of Hyp affects its intra- and intermolecular interactions and may play a critical role in its co-phase-separation with wild-type CTD, however our computational results suggest that the observed “effective” lower homotypic interactions of Hyp (*i.e.*, no co-phase-separation) mainly originate from the conformational restraint associated with the shift in the isomer populations. We made the following changes to the manuscript highlighting the work suggested by the reviewer.

Changes made in the manuscript:

(page 18/19)

We conjecture that the altered *cis-trans* isomerization within the hydroxyproline CTD variant modifies the propensity to form β -turns subsequently reducing the ability of CTD^{HYP} to undergo phase-separation. In line with this assertion, MD simulations showed that the *cis* conformer of both proline and hydroxyproline is crucial for driving the compactness of the CTD, whilst the extra hydroxyl group of hydroxyproline has only minor effect on CTD compactness. Unlike proline, hydroxyproline disfavors the *cis* conformation of the prolyl-peptidyl bond and has lower energy barrier for the *cis-trans* isomerization.⁵² Therefore, the different conformational properties of proline and hydroxyproline variants might be associated with the lack of co-mixing between mGFP-CTD^{HYP} and mCherry-CTD in our mixing phase-separation experiments (**Fig. 1f,g**). However, we note that other factors may govern co-phase-separation.^{84,85}

- The final section deals with the action of isomerases. The authors show how rather than catalytic activity, the direct interaction of isomerases with the proteins is responsible for the dissolution of condensates. We would expect that if the *cis-trans* isomerization is an important determinant of LLPS, the catalytically active isomerase would more avidly dissolve condensates by favoring the *trans* isomer in WT (like the Hyp variant which doesn't undergo LLPS), but the authors find the opposite.

Thank you for your insightful comment. We interpret the experiments with isomerases in terms of their canonical behavior, wherein they do not change the isomer populations but instead lower the interconversion barrier between isomers. Therefore, our experimental observation is consistent with the conformation-dependent behavior of proline isomers in relation to LLPS.

Overall, the combination of the above makes it hard to follow the main conclusions one should draw from the data and would benefit from more clarification regarding expectations vs. observations or a rewrite with milder language (e.g. saying that the data shows isomerization is important rather than saying isomerization is more important than interactions)

We followed the reviewer's suggestion and provided additional clarification, as well as toned down our statements about the role of isomerization throughout the manuscript.

5. In the results section, the simulation data seems to show certain trends that we find interesting and encourage the authors to comment on.

- In the case of the aromatic variants, the authors observe that Y to F mutations in the CG model lead to more dense droplets and more contacts as compared to Y (at higher concentrations). The experimental data shows a reduction in phase separation with Y to F, but the simulations are the opposite (saturation concentration for the F mutant would be lower than Y). The Y to F mutation has been studied quite extensively with numerous single-bead

per residue models adequately capturing the trend (<https://doi.org/10.1002/pro.4094>, <https://doi.org/10.1073/pnas.2111696118>, [10.1038/s43588-021-00155-3](https://doi.org/10.1038/s43588-021-00155-3)). We ask the authors to clarify whether the current CG model adequately captures this well-known trend. Further, we do not fully understand why the relative contacts between Y and F would be concentration dependent in the MD simulation.

The reviewer makes a very important point about the parameterization of aromatic residues in coarse-grained models. In particular, aromatic Y and F residues are challenging due to their planar nature and stacking interactions. Atomistic MD simulations ([10.1038/s43588-021-00155-3](https://doi.org/10.1038/s43588-021-00155-3)) and quantum mechanical calculations ([10.1021/ja049434a](https://doi.org/10.1021/ja049434a)) indicate that stacking (and possibly T-shaped) interactions may be more favorable for Y compared to F. Another commonly used source for the parameterization of coarse-grained models are hydrophobicity scales. However, different hydrophobicity scales have been proposed and Y and F residues do not have a conserved order. Together with the known importance of F interactions in protein stability ([10.1016/j.ijbiomac.2023.127207](https://doi.org/10.1016/j.ijbiomac.2023.127207)) and phase-separation ([10.1021/acs.jpcc.0c06288](https://doi.org/10.1021/acs.jpcc.0c06288)), this suggests that the difference in the interaction strength of these residues may be solvent and conformation dependent ([10.1021/ja0121639](https://doi.org/10.1021/ja0121639)). We are aware of the studies mentioned by the reviewer, however, we decided to use a more detailed and versatile parameterization, such as Martini3. Nevertheless, the relatively lower solubility of F compared to Y in Martini3 (with enhanced protein-water interactions), combined with the poor description of its stacking interactions, possibly led to F over-aggregation driven by hydrophobic effect.

In conclusion, we agree with the reviewers that the use of different physics-based models could lead to opposite behavior of the Y-to-F mutation, and we have included all the above considerations in a separate paragraph dedicated to a critical comparison between experimental and simulation results (see our response to **Reviewer 2, point 1**).

Changes made to the manuscript:

Notably, our condensed phase CG simulations revealed enhanced interactions of phenylalanine residues (**Supplementary Fig. 5c**), which led to more compact condensates for the CTD^{Y1F} variant, as indicated by the size distribution of the protein clusters (**Table S4**).

- In the investigations of the different Ser and Thr mutants on secondary structure, we find it interesting that the effects often seem to be on beta sheets not involving the mutated residue (sheet 2 for S2A for example) and ask that the authors comment/discuss why this may be.

We agree with the Reviewer that the behavior of the mentioned local turn is rather unexpected as it shows a significant change in the population ratio (above 10%) with respect to the wild-type CTD despite not involving any amino acid substitution. This behavior appears to be the result of interactions between residues of the first heptad (particularly S5) and the Y residue of the second heptad repeat (see below the map of interaction energies). Due to the limited size of the construct, Y1 in the first heptad is not subject to the same upstream intramolecular interactions and therefore can more freely interact with downstream residues, mitigating the expected expansion of turn 1. Therefore, we prefer not to speculate more about the S2A effects.

6. In the results section, when discussing the phosphorylation, the S2A mutant leads to an increase of almost 1.5 times. This seems to be the only mutant for which an increase is observed, however, it is currently not discussed in the manuscript. We ask that the authors discuss this in the results section and provide some insight into why this may be happening and potential implications.

The Geyer lab recently determined the crystal structure of the activated CDK7 complex without a CTD substrate (PMID: 38405971). They modeled the position of the substrate peptide P₃T₄S₅P₆S₇Y₁S₂ using the coordinates of the peptide PKTPKKA from the Cdk2/CycA/substrate complex structure (PDB ID: 3qhr; PMID: 21565702). In their model (see below), we have circled the expected positions of S2A substitutions within the CTD peptide. These positions are located in hydrophobic regions of the CDK7 complex. This suggests that S2A substitutions could introduce additional favorable hydrophobic interactions, potentially making S2A a more effective substrate in our *in vitro* assays. Following the reviewer's suggestion, we discuss this in the revised manuscript.

[REDACTED]

Figure adapted from Duster et al. (PMID: 38405971)

Changes in the manuscript:

(page14)

Additionally, we observed increased phosphorylation of the S2A variant compared to the WT. A recent crystal structure of the activated CDK7 complex with a modeled CTD peptide substrate⁶³, extrapolated from the Cdk2/CycA/substrate complex structure⁶⁴, suggests that the S2A substitution may introduce additional favorable hydrophobic interactions in the binding pocket, likely making S2A a more effective substrate in our *in vitro* assay.

7. In the results as well as the discussion section, the authors argue that the solvation of Ser in the “spacer” regions coupled with strong interactions among tyrosine residues together dictate aggregation vs. LLPS. We point the authors to the following experimental data (<https://doi.org/10.1021/bi00507a030>) that demonstrates the solvation of Tyr is more favorable than Ser. Therefore, this does not serve as an explanation for the differences between Tyr and Ser. We encourage the authors to rethink this model and propose or justify why favorable (but less than Tyr) solvation of Ser attenuates aggregation.

We have removed the term “favorable solvation” of serine residues in the revised manuscript.

Aside from the above, we have the following minor corrections to suggest.

1. Fig1C, the S7A mutant is incorrectly labeled as S5A

We thank the reviewer for spotting the typo. In **Fig. 1c**, the S7A variant is correctly labeled in the revised manuscript.

2. In the results section, line 132, the authors state that their investigation reveals that the tags do not affect phase separation. We suggest using “minimally effect” rather than “do not”.

We have adopted the suggested wording and included new experimental data to support this statement (see **Supplementary Fig. 1f-i**).

3. In the discussion, where the authors discuss the importance of Tyr in phase separation (line 341-344), we would like to point to authors to the following work highlighting the role of Tyr (10.1038/s43588-021-00155-3) as well the hydrogen bonding of Tyr with other amino acids in LLPS (<https://doi.org/10.1073/pnas.2000223117>).

We thank the reviewer for pointing us to the relevant work in the context of our discussion. We have included these references in the revised manuscript.